# Curvature-aware Graph Attention for PDEs on Manifolds

Yunfeng Liao [1]    Jiawen Guan [1]    Xiucheng Li [1]

## Abstract

Deep models have recently achieved remarkable performances in solving partial differential equations (PDEs). The previous methods are mostly focused on PDEs arising in Euclidean spaces with less emphasis on the general manifolds with rich geometry. Several proposals attempt to account for the geometry by exploiting the spatial coordinates but overlook the underlying intrinsic geometry of manifolds. In this paper, we propose a Curvature-aware Graph Attention for PDEs on manifolds by exploring the important intrinsic geometric quantities such as curvature and discrete gradient operator. It is realized via parallel transport and tensor field on manifolds. To accelerate computation, we present three curvature-oriented graph embedding approaches and derive closed-form parallel transport equations, and a subtree partition method is also developed to promote parameter-sharing. Our proposed curvature-aware attention can be used as a replacement for vanilla attention, and experiments show that it significantly improves the performance of the existing methods for solving PDEs on manifolds. Our code is available at https://github.com/Supradax/CurvGT.

## 1. Introduction

Partial differential equations (PDEs) serve as fundamental tools for describing a wide range of scientific phenomena. However, deriving closed-form solutions for general PDEs is often infeasible. The numerical methods such as finite difference methods and finite element methods, discretizing PDE domains into grids, offer a viable alternative but are often costly. Recently, data-driven deep models (Lu et al., 2021; Li et al., 2023d; 2020a; Kovachki et al., 2023) have emerged as powerful tools for PDE-governed tasks. They solve PDEs by learning the mapping between input-output pairs in an end-to-end manner. Thus they outperform numerical solvers in time-sensitive scenarios (e.g., real-time rendering in gaming engines) while slight inaccuracy is tolerable. However, most of them are dedicated to PDEs in Euclidean space. To generalize the methods to discrete manifolds, the work Geo-FNO (Li et al., 2023a) and Transolver (Wu et al., 2024) are proposed.

Geo-FNO (Li et al., 2023a) attempts to learn a universal mapping that transforms a curved surface into regularly spaced grids and then solves the PDEs in Euclidean space with Fourier neural operators. However, the reduction to Euclidean space might fail since the existence of such a universal parametrization is not always guaranteed. By contrast, Transolver (Wu et al., 2024) directly handles the ambient space in which the surface resides. By establishing a global coordinate system, it encodes relative positions into node features so as to distinguish different parts of the surface. Nevertheless, relying solely on spatial coordinates leaves the intrinsic geometry of the manifold unexplored. In particular, it is a well-established theory that the curvature of underlying manifolds dominates the physical laws of observable phenomena (Naber, 2011). Hence, the curvature is essential for developing an accurate solver for PDEs on the manifolds. However, neither Geo-FNO nor Transolver explicitly considers it in their model design.

On the other hand, the discrete manifolds are often represented by graphs (with node coordinates) and Graph Transformers (Ying et al., 2021; Zhang et al., 2023) have achieved promising results on a wide spectrum of graph-related tasks, whose successes can largely be attributed to the inductive bias injected into the self-attention to capture the underlying topology and geometry. However, as one of the most important geometry quantities of manifolds, the curvature is still under-explored. To further explore the intrinsic geometry of the underlying manifold, we propose a Curvature-aware Graph Attention for time-dependent PDEs on discrete manifolds, which naturally encodes the curvature into the self-attention mechanism. However, simply treating curvatures as additional node features has limited impact because it only reveals the local curvature at points, and it is the curvature of the path connecting points that dominates the physics laws. In addition, in contrast to message-passing in Euclidean space, the tangent vectors of different points

---

[1]School of Computer Science and Technology, Harbin Institute of Technology (Shenzhen), China. Correspondence to: Xiucheng Li <lixiucheng@hit.edu.cn>.

on a manifold residing in different tangent spaces own distinct local coordinates, which prevents us from aggregating information with a shared linear mapping.

To address these challenges, we propose to generalize the vanilla attention to manifolds via parallel transport. As a well-developed mathematical tool in differential geometry, parallel transport can naturally carry the path curvature from one point to another and align the local coordinate systems of different tangent spaces. Moreover, we propose to replace the linear mapping with the tensor field (multi-linear mapping) on manifolds to generalize the message aggregation. This is illustrated in Figure 1. To accelerate the computation of parallel transport, we present three curvature-oriented graph embedding methods and derive closed-form parallel transport equations accordingly; we also develop a subtree partition to promote parameter-sharing for the tensor field implementation. Lastly, a multi-head curvature-aware attention is presented to enhance the representation learning capability. Our proposed curvature-aware attention can be used as a replacement for vanilla attention in existing methods such as Graph Transformers, Graph Attention Networks, etc. The experimental results show that it is able to improve the performances of the existing methods significantly for PDEs on general manifolds. To summarize, our contributions are as follows.

- We propose a novel Curvature-aware Graph Attention for PDEs on manifolds, which naturally incorporates the curvature and discrete gradient operator as the inductive bias.

- The proposed attention is realized by a combination of parallel transport and tensor fields on manifolds; we present three curvature-oriented graph embedding methods and derive closed-form parallel transport equations to accelerate computation, and a subtree partition method is also proposed to promote parameter-sharing.

- Our proposed curvature-aware attention can be used as a direct replacement for self-attention. It significantly improves the performances of the existing methods for solving PDEs on manifolds.

## 2. Related Work

**Neural Operator.** Physics-Informed Neural Networks (PINN) are originally proposed to solve PDEs with exact formulas in continuous domains (Raissi, 2018), applied to fluid simulation in graphics (Jain et al., 2024), which is, however, intractable on a manifold. PINN based on graphs can better adapt to manifolds in the form of point clouds and structured meshes (Brandstetter et al., 2022). Different from PINNs, neural operators map a function to another by leveraging data-driven loss instead of the physical loss

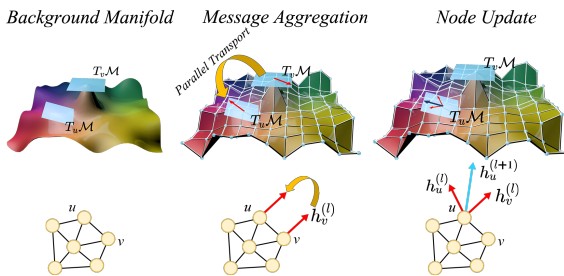

*Figure 1.* By interpreting node features as tangent vectors on the manifold, the message passing on a manifold is naturally defined as the parallel transport onward and node update can be generalized to multi-linear mapping in the tangent vector space.

given by PDEs (Lu et al., 2021; Jin et al., 2022; Li et al., 2023d). As for neural operators on graphs, spatial methods like multi-scale techniques implemented by pooling (Gao & Ji, 2019) or matrix decomposition (Li et al., 2020b) and spectral methods from Fourier operator (Li et al., 2020a; 2023a) to wavelet decomposition (Gupta et al., 2021; Tripura & Chakraborty, 2023; Xiao et al., 2023) are respectively proposed to enhance the graph-model performance. Spatial and spectral methods have recently achieved seamless integration. U-FNO (Wen et al., 2022) and U-NO (Rahman et al., 2023) utilize U-Net and FNO to capture multi-scale features in spatial and spectral domains, respectively. Since spectral methods are easier to handle in regular spaces, Geo-FNO (Li et al., 2023a) maps an irregular spatial domain to a regular grid followed by spectral blocks while LSM (Wu et al., 2023) first hierarchically projects a high-dimension spatial domain to latent physical tokens.

The paradigm of operator learning has been recently shifted to Transformer architectures with a focus on finding PDE-compatible attention mechanisms. HT-Net (Liu et al., 2023) adopts the canonical *V-cycle* (Gao & Ji, 2019; Li et al., 2020b) to obtain hierarchical attention on the PDE domain; GNOT (Hao et al., 2023) gains insight from Mixture of Experts(MoE) in which boundary shapes and edge features get involved in cross-attention; Transolver (Wu et al., 2024) decomposes the non-Euclidean PDE domain into slices and establishes slice-wise attention onward. To mitigate the quadratic attention problem in transformers, two types of softmax-free attention (Cao, 2021) are proposed to enhance the efficiency, which is further extended in OFormer (Li et al., 2023c) and FactFormer (Li et al., 2023e).

**Geometric Graph Embedding**. It is usually assumed that data features concentrate on a low-dimension manifold in a hidden feature space (Ghojogh et al., 2023). Among those non-trivial manifolds, hyperbolic space is a popular choice as the embedded space for hierarchical graphs, which is realized by bringing in gyrovector spaces (Peng et al., 2022) or classical hyperbolic models (Nickel & Kiela, 2017; 2018). In our task, the graph has a natural geometry inherited from

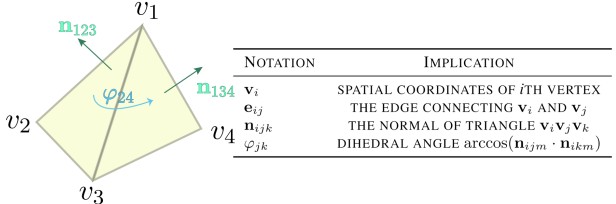

*Figure 2.* Notations of Geometric Quantities

| NOTATION | IMPLICATION |
|---|---|
| $\mathbf{v}_i$ | SPATIAL COORDINATES OF $i$TH VERTEX |
| $\mathbf{e}_{ij}$ | THE EDGE CONNECTING $\mathbf{v}_i$ AND $\mathbf{v}_j$ |
| $\mathbf{n}_{ijk}$ | THE NORMAL OF TRIANGLE $\mathbf{v}_i\mathbf{v}_j\mathbf{v}_k$ |
| $\varphi_{jk}$ | DIHEDRAL ANGLE $\arccos(\mathbf{n}_{ijm} \cdot \mathbf{n}_{ikm})$ |

the manifold $\mathcal{M}$ by discretization. The work that utilizes spatial coordinates as geometric information has been studied (Wu et al., 2024) whereas the intrinsic geometry of surfaces has not yet been explored.

## 3. Preliminaries

**Riemannian Manifold**. A Riemannian manifold $\mathcal{M}$ is a topological space that locally resembles Euclidean space and is equipped with a positive metric tensor field $g$. $\forall p \in \mathcal{M}$, there is an associated tangent space $T_p\mathcal{M}$ with inner product $g|_p$. The cotangent vector space $T_p^*\mathcal{M}$ is the dual space of $T_p\mathcal{M}$, consisting of linear functionals that map $\mathbf{u} \in T_p\mathcal{M}$ to a scalar via $\mathbf{v}^*(\mathbf{u}) := \langle \mathbf{v}, \mathbf{u} \rangle_{g|_p}$. A tensor is a multilinear mapping comprised of tangent vectors and cotangent vectors. For example, a (1,1)-tensor $\mathbf{u} \otimes \mathbf{v}^*$ acting on a tangent vector $\mathbf{w}$ gives a tangent vector $\langle \mathbf{v}, \mathbf{w} \rangle_{g|_p} \mathbf{u}$ while a (0,2)-tensor $\mathbf{u}^* \otimes \mathbf{v}^*$ acting on two tangent vectors $\mathbf{w}_1, \mathbf{w}_2$ in order gives a scalar $\langle \mathbf{u}, \mathbf{w}_1 \rangle_{g|_p} \langle \mathbf{v}, \mathbf{w}_2 \rangle_{g|_p}$.

**Parallel Transport**. Though Euclidean vectors can be paralleled in any manner without changing their inner products. However, to preserve inner products on $\mathcal{M}$, their components must vary in accordance to $g$ and the specific path $\gamma$ between $p, q \in \mathcal{M}$. Formally, let $\gamma(0) = p, \gamma(s) = q$, and the tangent vector $\mathbf{u} \in T_p\mathcal{M}$ is parallel transported along $\gamma$, resulting in the vector $\Gamma(\gamma)_0^s\mathbf{u} \in T_q\mathcal{M}$. The parallel transport of a cotangent vector $\mathbf{v}^*$ can be defined in consistency with the inner-product-preserving nature by: $\Gamma(\gamma)_0^s\mathbf{v}^*(\mathbf{u}) := \langle \Gamma(\gamma)_0^s\mathbf{v}, \mathbf{u} \rangle_{g|_q}$.

**Curvature**. Curvature is an essential intrinsic geometry property of manifolds and is central to many physics theories such as general relativity (Stephani, 2004), gauge theory (Naber, 2011), and string theory (Zwiebach, 2009). The sectional curvature $K(\mathbf{u}, \mathbf{v})|_p$ measures how much the geometry of a 2-dimensional sectional of manifold spanned by $\mathbf{u}, \mathbf{v} \in T_p\mathcal{M}$ deviates from being flat. For the 2-dimensional manifolds, the sectional curvature is identical to the Gaussian curvature $K$ that characterizes the degree and nature of local bending (Tu, 2017). The Gaussian curvature $K|_{\mathbf{v}_i}$ at point $\mathbf{v}_i$ can be estimated with the aid of the normal bundle theory (Yau & Gu, 2008) in a robust manner:

$$K|_{\mathbf{v}_i} = \frac{1}{2} \sum_{\mathbf{v}_j \in \mathcal{N}(\mathbf{v}_i)} \frac{\mathbf{v}_j - \mathbf{v}_i}{||\mathbf{v}_i - \mathbf{v}_j||_2} \varphi_{ij} \cdot \mathbf{n}. \quad (1)$$

The illustration of geometric quantities $\mathbf{n}, \mathbf{v}_i, \varphi_{ij}$ can be found in Figure 2.

## 4. Method

**Problem setup**. In this work, we consider the time-dependent PDEs on a manifold $\mathcal{M}$ of the form

$$\frac{du(\mathbf{x}, t)}{dt} = \mathcal{R}(u) \quad (2)$$

where $\mathbf{x} \in \mathcal{M}$, $u(\mathbf{x}, t)$ is a function of $\mathbf{x}$ and time $t$, $\mathcal{R}$ is a linear or nonlinear differential operator. For instance, $\mathcal{R}(u) = \Delta u(\mathbf{x}, t) + f(\mathbf{x}, t)$ for a heat equation, $u(\mathbf{x}, t)$ and $f(\mathbf{x}, t)$ represent the observed temperature field and heat-source distribution, respectively. In practice, $\mathcal{M}$ is often given as a discrete manifold, i.e., a graph $G$ with node set $V$ and edge $E$. Given a discrete manifold $G$ and a collection of $m$ functions $u_1(\mathbf{x}, t), u_2(\mathbf{x}, t), \ldots, u_m(\mathbf{x}, t)$ on $G$ at time $t$, the neural operator $\mathcal{F}$ aims to produce the function $u(\mathbf{x}, t + 1)$ on $G$ at $t + 1$. As most of the PDEs of interest in practice are established on 2-dimensional manifolds, we will primarily focus on these manifolds in this study. $u_i(\mathbf{x}, t)$ will also be denoted by $u_i^{(t)}$ for short.

### 4.1. Design Motivation and Challenges

**Design Motivation**. Self-attention mechanisms (Vaswani et al., 2017) have become the de-facto choice for foundation model designs across various domains. In particular, Graph Transformers (Ying et al., 2021; Zhang et al., 2023) have extended their successes to graph structure data and achieved state-of-the-art performance on a wide range of graph-related tasks. The success of Graph Transformer can be largely credited to the inductive bias injected into the self-attention to capture the underlying local structures, which are oblivious to the naive implementation. For example, Graphormer (Ying et al., 2021) introduces the node centrality and shortest path distance (SPD) to correct the attention score, thereby making it aware of the node degree and pairwise node relationship; Graphormer-GD (Zhang et al., 2023) further explores the Resistance Distance (RD) to enhance its expressiveness. In a nutshell, the core idea behind this success is to introduce bias to help the attention mechanism be aware of the local topology (node degree) and geometry (pairwise node distance) of the underlying graph.

Inspired by the above observation, it is appealing to generalize such success to the discrete manifold, represented as a graph with rich geometry. As mentioned in Section 1, one of the critically important geometry quantities on the manifold that dominate the underlying physics law is the curvature. In this paper, we aim to propose a curvature-aware self-attention mechanism to solve the PDEs on a manifold. Instead of adopting the Graph Transformer architecture, we still opt for the message-passing framework to avoid the

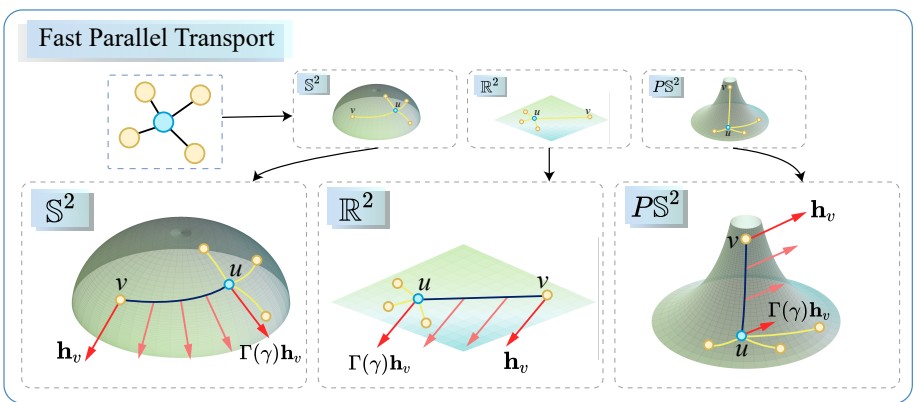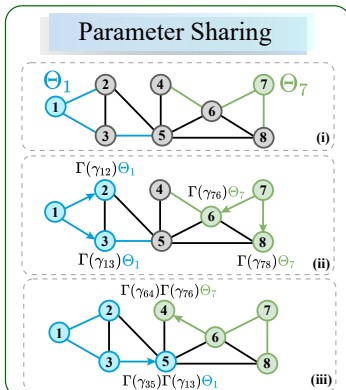

*Figure 3.* A node is locally embedded into one of the constant Gaussian curvature surfaces, enabling us to perform fast parallel transport. Parameter sharing are implemented by broadcasting source node parameters via parallel transport.

quadratic complexity $O(|V|^2)$.

**Challenges of Encoding Curvature**. To illustrate our method, we first recall the general graph message-passing steps by focusing on one single-layer transformation,

$$\mathbf{m}'_u = \sum_{v \in \mathcal{N}(u)} \alpha_{uv} \mathbf{W} \mathbf{h}_v \qquad (3)$$

$$\mathbf{h}'_u = \sigma \left( \tilde{\mathbf{W}} \mathbf{h}_u + \mathbf{m}'_u \right) \qquad (4)$$

$$\alpha_{uv} = \text{softmax}_v(\mathbf{a}^\top [\mathbf{h}_u \| \mathbf{h}_v]) \qquad (5)$$

in which $\mathbf{h}_v \in \mathbb{R}^D, \mathbf{h}'_u \in \mathbb{R}^{D'}$ are node representations for node $u, v \in V$, $\mathbf{m}'_u$ is the aggregated message for node $u$ from its neighbors $v$, whereas $\mathbf{W}, \tilde{\mathbf{W}} \in \mathbb{R}^{D' \times D}, \mathbf{a} \in \mathbb{R}^{1 \times 2D}$ are learnable parameters, and $\sigma()$ is the activation function. To incorporate curvature information, one may attempt to mimic the Graphormer by either computing the local curvature of node $u, v$ denoted by $\mathbf{c}_u, \mathbf{c}_v$ and concatenating it to $\mathbf{h}_u, \mathbf{h}_v$ or adding a bias term $\phi_{uv}$ (e.g., SPD or RD) to correct the attention score, i.e., replacing Eq. (5) with

$$\alpha_{uv} = \text{softmax}_v(\mathbf{a}^\top [\mathbf{h}_u \| \mathbf{c}_u \| \mathbf{h}_v \| \mathbf{c}_v]) + \phi_{uv}.$$

However, such a straightforward adaptation will not work for two reasons. (i) It is the curvature of the path between $u, v$ that dominates the physics law, simply concatenating $\mathbf{c}_v$ to $\mathbf{h}_v$ will not work as $\mathbf{c}_v$ can only reveal the local curvature at point $v$; unlike the distance (SPD or RD), there is no closed-form formula to calculate the curvature of the path between two points $u, v$ on a manifold. (ii) To acquire the aggregated message $\mathbf{m}'_u$ in Eq. (3), we transform $\mathbf{h}_v$ from $\mathbb{R}^D$ to $\mathbb{R}^{D'}$ with a linear mapping $\mathbf{h}_v \mapsto \mathbf{W} \mathbf{h}_v$. This is feasible as both $\mathbf{h}_v$ and $\mathbf{m}'_u$ live in the Euclidean space. However, on a manifold $\mathcal{M}$, $\mathbf{h}_v$ and $\mathbf{m}'_u$ will reside in two different tangent spaces, $T_v\mathcal{M}$ and $T_u\mathcal{M}$, respectively, and it makes no sense to transform vectors via linear mapping between two tangent spaces since each of them owns a distinct local coordinate. Hence, it is unclear how to aggregate

the representations from different tangent spaces $T_v\mathcal{M}$ with $v \in \mathcal{N}(u)$ to produce $\mathbf{m}'_u$. For the same reason, Eq. (4) and Eq. (5) have similar issue.

To address the above two challenges, we propose to align different tangent spaces on a manifold via parallel transport and generalize the matrix multiplication (linear mapping) with the tensor field (multi-linear mapping). By integrating the two techniques in a systematic way, we develop a curvature-aware graph attention mechanism that can more effectively solve the PDEs on manifold, the details are presented in Section 4.2. Besides, to further accelerate the computation of parallel transport, we propose three curvature-oriented graph embedding methods and derive closed-form parallel transport equations in Section 4.3, and in Section 4.4 we also present a subtree partition approach to promote the parameter-sharing for the tensor field implementation. The framework is presented in Figure 3.

### 4.2. Curvature-aware Graph Attention

The parallel transport serves as a foundation tool for our proposed curvature-aware graph attention. To compute $\mathbf{m}'_u$, each neighboring node representation $\mathbf{h}_v$ is moved from $T_v\mathcal{M}$ to $\Gamma(\gamma)_0^s[\mathbf{h}_v] \in T_u\mathcal{M}$ via parallel transport. The benefits are twofold: 1) the parallel transport is a well-developed tool in differential geometry and it can reveal the curvature of the space (more concretely, the curve along which it moves), and thus we can encode the curvature of the path $\gamma$ into the transported vector $\Gamma(\gamma)_0^s[\mathbf{h}_v]$; 2) the transported $\Gamma(\gamma)_0^s[\mathbf{h}_v]$ becomes a tangent vector in $T_u\mathcal{M}$, which enables us to generalize the matrix multiplication with the tensor field to calculate $\mathbf{m}'_u$.

To motivate our design, we first rewrite the matrix multiplication with its singular value decomposition form:

$$\mathbf{Wh} = \sum_i \lambda_i \mathbf{p}_i \mathbf{q}_i^\top \mathbf{h} = \sum_i \lambda_i \mathbf{p}_i \langle \mathbf{q}_i, \mathbf{h} \rangle$$
$$= \sum_i \lambda_i (\mathbf{p}_i \otimes \mathbf{q}_i^*)(\mathbf{h}) \tag{6}$$

in which $\mathbf{p}_i, \mathbf{q}_i$ are left- and right-singular vectors of $\mathbf{W}$. The key observation is that the inner product $\langle \mathbf{q}_i, \mathbf{h} \rangle$ times $\mathbf{p}_i$ can be viewed as a $(1,1)$-tensor or multi-linear mapping $\mathbf{p}_i \otimes \mathbf{q}_i^*$ acting on a tangent vector $\mathbf{h}$, which can be generalized to manifolds naturally. In view of tensor fields, we can rewrite the first two message-passing steps in Eq. (3) and Eq. (4) in the following forms.

$$\mathbf{m}'_u = \sum_{v \in \mathcal{N}(u)} \alpha_{uv} (\mathbf{w}_1 \otimes \mathbf{w}_2^*)(\Gamma(\gamma)_0^s[\mathbf{h}_v]) \tag{7}$$

$$\mathbf{h}'_u = \sigma \left( (\tilde{\mathbf{w}}_1 \otimes \tilde{\mathbf{w}}_2^*)(\mathbf{h}_u) + \mathbf{m}'_u \right) \tag{8}$$

in which $\mathbf{w}_1, \mathbf{w}_2, \tilde{\mathbf{w}}_1, \tilde{\mathbf{w}}_2 \in \mathbb{R}^3$ are parameters to learn. We first move $\mathbf{h}_v \in T_v \mathcal{M}$ via parallel transport to obtain a tangent vector $\Gamma(\gamma)_0^s[\mathbf{h}_v] \in T_u \mathcal{M}$. Then we apply a $(1,1)$-tensor $\mathbf{w}_1^* \otimes \mathbf{w}_2$ to $\Gamma(\gamma)_0^s[\mathbf{h}_v]$ to yield a transformed tangent vector in $T_u \mathcal{M}$, and consequently, $\mathbf{m}'_u$ will also be a tangent vector in $T_u \mathcal{M}$, which can be used to generate $\mathbf{h}'_u$. In a similar spirit, Eq. (5) can be generalized with a $(0,2)$-tensor $\mathbf{a}_1^* \otimes \mathbf{a}_2^*$ to produce a scalar as follows:

$$\alpha_{uv} = \mathrm{softmax}_v \left( (\mathbf{a}_1^* \otimes \mathbf{a}_2^*)(\mathbf{h}_u, \Gamma(\gamma)_0^s[\mathbf{h}_v]) \right) \tag{9}$$

where $\mathbf{a}_1, \mathbf{a}_2 \in \mathbb{R}^3$ are learnable parameters.

**Implementation of Tensor Field**. Our proposed curvature-aware graph attention involves a $(1,1)$-tensor $\mathbf{w}_1 \otimes \mathbf{w}_2^*$ and a $(2,0)$-tensor $\mathbf{a}_1^* \otimes \mathbf{a}_2^*$ operation, which is defined by inner product on the manifold with associated metric tensor field $g$ as follows (assuming both tensors are assigned to point $u$ and $\mathbf{h}, \mathbf{h}' \in T_u \mathcal{M}$):

$$(\mathbf{w}_1 \otimes \mathbf{w}_2^*)(\mathbf{h}) = \mathbf{w}_1 \langle \mathbf{w}_2, \mathbf{h} \rangle_{g|_u},$$
$$(\mathbf{a}_1^* \otimes \mathbf{a}_2^*)(\mathbf{h}, \mathbf{h}') = \langle \mathbf{a}_1, \mathbf{h} \rangle_{g|_u} \langle \mathbf{a}_2, \mathbf{h}' \rangle_{g|_u}. \tag{10}$$

By Jacobi field theory (Jost, 2017), a trivial Jacobi field along a radial geodesic enables a Taylor expansion to estimate the metric tensor field $g$ by curvature tensor $R_{iklj}$:

$$g_{ij}(x) = \delta_{ij} - \frac{1}{3} R_{iklj} x^k x^l + O(|x|^3) \tag{11}$$

in which $R_{iklj}$ is tractable with Taubin tensor (Taubin, 1995). Notably, for a surface embedded in $\mathbb{R}^3$, there is only one independent component $R_{1212}$, which simplifies computation dramatically.

**Remark 1**. As a side benefit, generalizing the matrix multiplication in Euclidean space to the tensor field on manifold $\mathcal{M}$ ensures that the obtained $\mathbf{h}'_u$ in Eq. (8) still resides in $T_u \mathcal{M}$, which permits us to stack up multiple curvature-aware graph attention layers to learn more expressive representations.

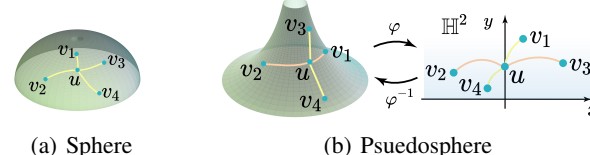

(a) Sphere    (b) Psuedosphere

*Figure 4.* Subgraphs are embedded into constant curvature surfaces. It is common to handle the pseudosphere $P\mathbb{S}^2$ with the Poincaré's half-plane $\mathbb{H}^2$ via isometry $\varphi : P\mathbb{S}^2 \to \mathbb{H}^2$.

**Remark 2**. In our proposed attention, message passing is achieved by using tangent vector fields. However, in many PDEs, both input and output are scalar fields. The conversion from a vector field to a scalar field can be achieved by adding a learnable cotangent vector field. Conversely, the most natural vector field related to a scalar field is its gradient, obtained by the discrete gradient operator $G$ (Jacobson, 2013), which is also an important geometry quantity.

### 4.3. Fast Parallel Transport

To support our proposed curvature-aware graph attention, we need a fast and accurate implementation of parallel transport on manifold $\mathcal{M}$. Since $\mathcal{M}$ is often represented by its discrete version $G = (V, E)$, it suffices to consider the design of parallel transport along a directed edge $(v, u) \in E$. To this end, we propose to embed an edge $(v, u)$ into three types of constant Gaussian curvature surfaces, namely, the sphere $\mathbb{S}^2$, the plane $\mathbb{R}^2$, and the pseudosphere $P\mathbb{S}^2$, which owns positive, zero, and negative curvature, respectively. The idea is to approximate the local geometry of $(v, u)$ with the three constant curvature surfaces depending on the value of curvature $K|_u$ at $u$ as follows:

$$\begin{cases} \mathbb{S}^2 & \text{if } K|_u > \epsilon, \\ \mathbb{R}^2 & \text{if } -\epsilon \le K|_u \le \epsilon, \\ P\mathbb{S}^2 & \text{if } K|_u < -\epsilon, \end{cases} \tag{12}$$

in which threshold $\epsilon > 0$ is given. Therefore, we only need to consider the parallel transport along the three constant Gaussian curvature surfaces. To offer a fast implementation, we will derive a closed-form formula for each of them in the remainder of this section. First, the parallel transport on $\mathbb{R}^2$ is simply an identity map. Next, we consider $\mathbb{S}^2$ and $P\mathbb{S}^2$.

**Parallel Transport on $\mathbb{S}^2$**. The embedding of edge $(v, u)$ into $\mathbb{S}^2$ is simple. We can just treat the spatial coordinates $\mathbf{u}, \mathbf{v} \in \mathbb{R}^3$ of $u, v$ as two vectors on $\mathbb{S}^2$ and calculate their rotation angle $\theta$, which is closely related to the path curvature from $v$ to $u$ on $\mathbb{S}^2$. Theorem 4.1 states that the parallel transport on the sphere is uniquely determined by $\theta$.

**Theorem 4.1** (Parallel transport on $\mathbb{S}^2$). *Let $\mathbf{u}, \mathbf{v} \in \mathbb{S}^2 \subset \mathbb{R}^3$, $\theta := \arccos(\mathbf{u} \cdot \mathbf{v})$ and the orthonormal basis be $\{\mathbf{e}_1 = \mathbf{u}, \mathbf{e}_2 = \mathbf{u} \times \mathbf{v} / \|\mathbf{u} \times \mathbf{v}\|_2, \mathbf{e}_3 = \mathbf{e}_1 \times \mathbf{e}_2\}$. Suppose the geodesic between $\mathbf{v}$ and $\mathbf{u}$ is $\gamma(t) = \cos\theta \mathbf{e}_1 + \sin\theta \mathbf{e}_3$,*

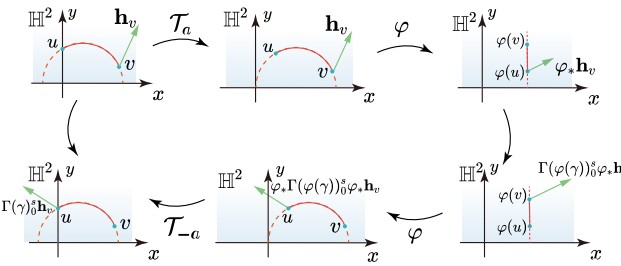

Figure 5. The computation diagram commutes.

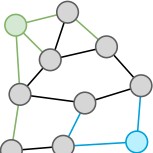
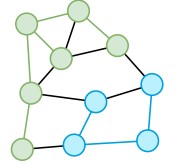

Initial State     Parameter Sharing Scheme

Figure 6. The graph is partitioned into two subtrees with depth $d = 3$, highlighted in different colors.

then $\mathbf{h}_v = a\mathbf{e}_3 + b\mathbf{e}_2 \in T_v\mathbb{S}^2$ is parallel transported to $\Gamma(\gamma)_0^\theta \mathbf{h}_v = a\cos\theta \mathbf{e}_3 - a\sin\theta \mathbf{e}_1 + b\mathbf{e}_2 \in T_u\mathbb{S}^2$.

The proof of the theorem is given in Appendix A. In short, Theorem 4.1 offers us a very cheap manner to transport $\mathbf{h}_v$ from the tangent space $T_v\mathbb{S}^2$ to $T_u\mathbb{S}^2$ on the sphere.

**Parallel Transport on** $P\mathbb{S}^2$. To derive the parallel transport in $P\mathbb{S}^2$, we propose to map an edge $(v, u)$ to Poincaré's half-plane $\mathbb{H}^2$ with an isometry $\varphi : P\mathbb{S}^2 \to \mathbb{H}^2$, because the two spaces are isometric and $\mathbb{H}^2$ is a well-studied hyperbolic space that enjoys appealing properties. To achieve the transformation, it suffices to design the pushforward $\varphi_* : T_u P\mathbb{S}^2 \to T_p\mathbb{H}^2$ of the isometry by making it preserve the direction and distance between points in $P\mathbb{S}^2$. To this end, we note that the two principal directions at a point $u$ are exactly the two axes of $\mathbb{H}^2$, as illustrated in Figure 4-(b). Hence, $\varphi_*$ can be designed to map $\mathbf{u}$ to the coordinate $(0, 1)$ in $\mathbb{H}^2$ and $\mathbf{v}$ to the location along the direction $\text{proj}_{\mathbf{n}_u^\perp}(\mathbf{v} - \mathbf{u})$ with a distance $d_{\mathbb{R}^3}(\mathbf{u}, \mathbf{v})$, where $\text{proj}$ is the projection operator and $\mathbf{n}_u^\perp$ denotes the orthogonal complement of normal vector at $u$ (the tangent plane of $P\mathbb{S}^2$ at point $u$). In the remainder of this section, we will denote the coordinate of $v \in \mathbb{H}^2$ by $(x_v, y_v) \in \mathbb{R}^2$ or represented by the complex number $z_v = x_v + iy_v$.

Now we are ready to derive the parallel transport in $\mathbb{H}^2$. First, if $\gamma$ happens to be a line segment parallel to $y$-axis, $\Gamma(\gamma)_0^s$ is then a simple scaling transformation as stated by Theorem 4.2. However, in general, $\gamma$ is more likely to be a section of semi-circle centered on $x$-axis and Theorem 4.2 will fail. Fortunately, we can turn the semi-circle into a segment by inversion $z \mapsto 1/z$ and translation $\mathcal{T}_a : z \mapsto z + a$, which are isometric transformations in $\mathbb{H}^2$ (as justified by Lemma 4.3). Meanwhile, Lemma 4.4 states that parallel transport and the pushforward of isometric transformation are commutative (the lemma is illustrated in Figure 5), enabling us to transform the semi-circle into a segment parallel $y$-axis and then apply Theorem 4.2 to derive the closed-form parallel transport on $\mathbb{H}^2$ for the general cases. The result is summarized in Theorem 4.5, the computation only involves $2 \times 2$-matrix multiplication, which incurs little overhead.

**Theorem 4.2** (Parallel transport on $\mathbb{H}^2$). *Let $u, v \in \mathbb{H}^2$ s.t. $x_u = x_v$. Parallel transport along the segment from $v$ to $u$ is given by $\Gamma(\gamma)_0^s\mathbf{h}_v = y_u/y_v\mathbf{h}_v \in T_u\mathbb{H}^2$.*

**Lemma 4.3.** *The special Möbius group $M(\mathbb{C}) := \{z \mapsto (cz + d)/(az + b) : a, b, c, d \in \mathbb{C}, ad - bc = 1\}$ is an isometric transformation group of $\mathbb{H}^2$. Inversion $z \mapsto 1/z$ is a special case of $M(\mathbb{C})$ which maps circles that pass through the origin to a line parallel to the $y$-axis and vice versa. Besides, the translations $\mathcal{T}_a(a \in \mathbb{R})$ along $x$-axis in $\mathbb{H}^2$ also form an isometric automorphism group.*

**Lemma 4.4.** *If $\varphi : (\mathcal{M}, g) \to (\mathcal{N}, h)$ is isometric, that is, $\langle \mathbf{u}, \mathbf{v}\rangle_{g|_p} = \langle \varphi_*\mathbf{u}, \varphi_*\mathbf{v}\rangle_{h|_{\varphi(p)}}$ holds for all $\mathbf{u}, \mathbf{v} \in T_p\mathcal{M}$, then the pushforward $\varphi_*$ and parallel transport commute:*

$$\varphi_* \circ \Gamma(\gamma)_0^s = \Gamma(\varphi(\gamma))_0^s \circ \varphi_*. \quad (13)$$

*Furthermore, if $\varphi \circ \varphi = \text{id}$ then we also have*

$$\Gamma(\gamma)_0^s = \varphi_* \circ \Gamma(\varphi(\gamma))_0^s \circ \varphi_*. \quad (14)$$

**Theorem 4.5** (Parallel transport on $\mathbb{H}^2$). *If $\gamma$ is on a semi-circle then by the commutativity between the pushforward of isometric transformation and parallel transport we have*

$$\Gamma(\gamma)_0^s\mathbf{h}_v = (\mathcal{T}_{-a})_* \circ \varphi_* \circ \Gamma(\varphi(\gamma))_0^s \circ \varphi_* \circ (\mathcal{T}_a)_*(\mathbf{h}_v). \quad (15)$$

*Moreover, let $(-a, 0)$ be the left intersection between $\gamma$ and $x$-axis, and $\tilde{z} := \varphi(z) = -1/(z+a)$, then Eq. (15) becomes*

$$\Gamma(\gamma)_0^s\mathbf{h}_v = \frac{\tilde{y}_u}{\tilde{y}_v}\mathbf{J}|_{\tilde{z}_u}\mathbf{J}|_{z_v+a}\mathbf{h}_v \in T_u\mathbb{H}^2 \quad (16)$$

*where $\mathbf{J}|_z = |z|^{-2}\begin{pmatrix} x^2 - y^2 & 2xy \\ -2xy & x^2 - y^2 \end{pmatrix}$.*

In our implementation, $\mathbf{h}_v$ is represented by a 3D-vector (for the convenience of computation) of dimension two (in its local coordinate). Hence, it can be transformed by a $2 \times 2$ matrix locally.

### 4.4. Subtree Partition to Facilitate Parameter Sharing

When generalizing the matrix multiplication from the Euclidean space to the tensor field on manifolds, the learnable

parameters $\Theta = \{\mathbf{w}_i, \tilde{\mathbf{w}}_i, \mathbf{a}_i\}$ $(i = 1, 2)$ in Eq. (7-9) will all fall in the tangent space $T_u\mathcal{M}$. In contrast to the Euclidean space $\mathbb{R}^n$, each $T_u\mathcal{M}$ is a local coordinate and it is geometrically unmeaningful to share parameters across distinct coordinates. As a consequence, we have to assign a group of different parameters to each node $u \in V$ and the number of parameters will grow with $|V|$, which hinders the statistical strength sharing across nodes and renders the learning infeasible.

To control the number of parameters and promote statistical strength sharing, we further develop a subtree partition approach. We start by noting that the tensor field is eventually instantiated with the inner product on manifolds in Eq. (10) and the parallel transport preserves inner products, i.e.,

$$\langle \Gamma(\gamma)_0^s[\mathbf{v}_1], \Gamma(\gamma)_0^s[\mathbf{v}_2]\rangle_{g|_v} = \langle \mathbf{v}_1, \mathbf{v}_2\rangle_{g|_u}. \quad (17)$$

This enlightens us to share parameters across nodes with parallel transport. Specifically, we can move the parameters $\Theta$ from a node $u$ to node $w$ via parallel transport and let them share one set of parameters. Then the curvature-aware attention becomes

$$\begin{aligned}
\mathbf{m}'_w &= \sum_{v \in \mathcal{N}(w)} \alpha_{wv} \Gamma(\eta)_0^t[\mathbf{w}_1^* \otimes \mathbf{w}_2](\Gamma(\gamma)_0^s[\mathbf{h}_v]) \\
\mathbf{h}'_w &= \sigma\left(\Gamma(\eta)_0^t[\tilde{\mathbf{w}}_1^* \otimes \tilde{\mathbf{w}}_2](\mathbf{h}_u) + \mathbf{m}'_w\right) \\
\alpha_{wv} &= \mathrm{softmax}_v\left(\Gamma(\eta)_0^t[\mathbf{a}_1^* \otimes \mathbf{a}_2^*](\mathbf{h}_w, \Gamma(\gamma)_0^s[\mathbf{h}_v])\right)
\end{aligned} \quad (18)$$

in which $\eta$ is the piecewise geodesic from $u$ to $w$ (to transport parameters $\Theta$) and $\gamma$ is the geodesic from $v$ to $w$ (to transport representation $\mathbf{h}_v$).

To reduce the average transporting distance, we start with a randomly selected node $u \in V$ (as the source node) and run the BFS (breadth first search) with a maximum depth $d$ to form a subtree $T$ of $G$. Next, we randomly select another source node $u'$ from the unvisited to repeat the above procedure until all nodes in $G$ are visited. In such a manner, we partition $G$ into a collection of subtrees. We only assign parameters $\Theta$ to source nodes and the other nodes in a subtree share the same set of parameters with the source nodes via parallel transport. Figure 6 shows a partition with two subtrees.

**Computation Complexity**. The complexity of a typical single-head GAT layer is $O(|V|D+|E|D)$ (Veličković et al., 2018) where $D$ is the number of features. Note that a set of edge-wise parameters to support parallel transport only includes a 2-order transform matrix, a 2-order metric matrix, and local coordinates while the parallel transport along each edge can be done within $O(1)$. A subtree with maximum depth $d$ requires parallel transport for $(d-1)$ times to ensure the parameters are broadcast to all nodes. Therefore, the complexity of a single-head layer is $O(|V|Dd + |E|D)$.

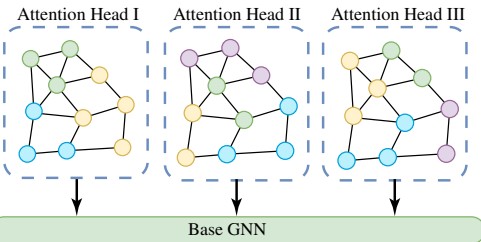

*Figure 7.* Multi-head attention via diverse subtree partitions.

### 4.5. Generalized Geometric Multi-Head Attention

Multi-head attention expects the model to learn from different represent subspaces (Vaswani et al., 2017). Following the same spirit and noting that the subtree partition is not unique, we present a multi-head attention mechanism on manifolds by different partitions, as shown in Figure 7. The realization is simple since it only involves an extra concatenation operation. Specifically, the curvature-aware graph attention layer with $C$ heads is shown as follows:

$$\begin{aligned}
\mathbf{m}'_{v \to w, i} &= \Gamma(\eta)_0^t[\mathbf{w}_{1,i}^* \otimes \mathbf{w}_{2,i}](\Gamma(\gamma)_0^s[\mathbf{h}_{v,i}]) \\
\mathbf{m}'_w &= \Big\|_{i=1}^C \sum_{v \in \mathcal{N}(w)} \alpha_{wv,i}\mathbf{m}'_{v \to w, i} \\
\mathbf{h}'_{w,i} &= \Gamma(\eta)_0^t[\tilde{\mathbf{w}}_{1,i}^* \otimes \tilde{\mathbf{w}}_{2,i}](\mathbf{h}_{w,i}) \\
\mathbf{h}'_w &= \sigma\left(\Big\|_{i=1}^C \mathbf{h}'_{w,i} + \mathbf{m}'_w\right) \\
\alpha_{wv,i} &= \mathrm{softmax}_v \Gamma(\eta)_0^t[\mathbf{a}_{1,i}^* \otimes \mathbf{a}_{2,i}^*](\mathbf{h}_{w,i}, \Gamma(\gamma)_0^s[\mathbf{h}_{v,i}])
\end{aligned} \quad (19)$$

where $\mathbf{h}_{w,i}$ is the feature tangent vector in $T_w\mathcal{M}$ of the $i$-th head and $\mathbf{m}_w, \mathbf{h}_w \in \mathbb{R}^{3C}$.

## 5. Experiments

**Experiment Settings**. A dataset is a collection of tuples $(u_1^{(t)}, ..., u_m^{(t)}; u^{(t+1)})$ and the associated neural operator is $\mathcal{F}: L^2(\mathcal{M}) \times ... \times L^2(\mathcal{M}) \to L^2(\mathcal{M})$. Following the paradigm in (Li et al., 2020a), we assume the input functions $\mathcal{I}$ following a certain distribution $\mu$ and define the loss by:

$$\mathcal{L} := \mathbb{E}_{\mathcal{I} \sim \mu} \ell(\mathcal{F}(\mathcal{I}), u^{(t+1)}) \quad (20)$$

where $\ell$ is a function that measures the difference between the output $\mathcal{F}(\mathcal{I})$ and the ground truth $u^{(t+1)}$. $\ell$ is usually chosen to be $L^p$ norm when solving PDEs in Euclidean spaces. Here we adopt $L^2$ norm $||\cdot||_2$ and Hilbert $H^1$ norm $||\cdot||_{H^1(\mathcal{M})}$ since they can be naturally extended to a general manifold $\mathcal{M}$:

$$||u(x)||_2^2 := \int_{x \in \mathcal{M}} u^2(x)dx, \quad (21)$$

$$||u(x)||_{H^1(\mathcal{M})}^2 := ||u(x)||_2^2 + ||\nabla u(x)||_2^2. \quad (22)$$

*Table 1.* Performances of different methods on the discrete wrinkle (see Figure 10) governed by isotropic diffusion, $p$-Laplacian diffusion and wave equation (see Appendix B.1), respectively. The relative errors of $L^2$ and $H^1$ are reported.

| Model | Isotropic Diffusion | | $p$-Laplacian Diffusion | | Wave Equation | |
|---|---|---|---|---|---|---|
| | $L^2(\%)$ | $H^1(\%)$ | $L^2(\%)$ | $H^1(\%)$ | $L^2(\%)$ | $H^1(\%)$ |
| ResNet | 2.008±2.187 | 7.470±9.149 | 29.362±6.412 | 54.117±12.384 | 4.218±3.917 | 4.484±4.325 |
| GCN | 7.947±0.805 | 44.587±11.203 | 5.445±2.885 | 12.923±8.527 | 21.783±1.586 | 24.901±1.985 |
| DeepONet | 2.296±0.046 | 8.977±0.534 | 37.855±0.530 | 48.302±2.134 | 25.773±1.051 | 27.536±1.399 |
| GraphTransformer | 0.061±0.018 | 0.221±0.056 | 9.695±4.236 | 14.443±4.780 | 11.848±4.495 | 13.679±5.439 |
| FourierType | 0.086±0.027 | 0.327±0.114 | 14.044±4.405 | 20.313±6.184 | 15.824±4.552 | 17.530±5.957 |
| GalerkinType | 0.077±0.028 | 0.291±0.107 | 13.456±2.711 | 16.928±2.938 | 14.204±5.655 | 15.748±7.033 |
| Graph U-Net | 7.043±0.282 | 41.989±6.142 | 10.971±1.912 | 49.765±32.059 | 16.184±0.697 | 19.116±2.217 |
| MKGN | 0.640±0.693 | 2.139±2.212 | 7.418±0.731 | 21.064±5.351 | 2.966±1.599 | 4.105±2.324 |
| GNOT | 0.040±0.023 | 0.160±0.061 | 2.466±1.453 | 4.390±1.923 | 0.351±0.181 | 0.457±0.274 |
| GINO | 0.320±0.011 | 1.415±0.051 | 42.819±1.149 | 52.579±1.011 | 2.043±0.555 | 2.136±0.343 |
| Geo-FNO | 0.075±0.029 | 0.249±0.055 | 1.936±0.389 | 8.177±1.375 | 2.394±0.245 | 2.196±0.306 |
| Transolver | 0.681±0.181 | 1.687±0.344 | 1.935±0.318 | 3.905±0.550 | 1.646±0.493 | 1.855±0.497 |
| Curv-GT (**Ours**) | **0.025±0.008** | **0.101±0.033** | **1.577±0.839** | **3.212±1.432** | **0.035±0.014** | **0.039±0.017** |

To calculate the above integrals, we first compute the mass matrix $\mathbf{M}$ with finite element methods. $\mathbf{M}$ is a diagonal matrix whose diagonal element $m_i$ is the surface area that vertex $v_i$ takes up. Therefore, we have $\|u(x)\|_2^2 \approx \mathbf{u}^\top \mathbf{M} \mathbf{u}$. Likewise, one can adopt a discrete gradient operator $G$ induced by a Witney basis to take the place of $\nabla$ (Jacobson, 2013), which is indeed a matrix:

$$||\nabla u(x)||_2^2 \approx \sum_{i=1}^{|V|} m_i \langle f(G\mathbf{u}), f(G\mathbf{u})\rangle_{g|_{v_i}} \quad (23)$$

where $f$ maps the vector field on faces to that on vertices and the inner product in each tangent space at $v_i$ are defined as those in constant curvature spaces.

**Datasets.** The previous methods (Li et al., 2023a) select a function $u(x,t)$ on a geometry object and compute its source terms $f(x,t)$ by the corresponding equations to obtain a data tuple $(u^{(t)}, f^{(t)}; u^{(t+1)})$. Such an approach is out of the following considerations: (i) Popular datasets (Li et al., 2023a) are limited to Euclidean spaces, which does not match the task we focus on; (ii) Convergence and accuracy of numerical methods on a general manifold cannot be guaranteed. Hence, we generate the dataset by selecting a collection of functions on a certain parametric surface in advance, and closed-form computations on parametric manifolds make the dataset more reliable. In this paper, various time-dependent PDEs on different manifolds are studied. More details are available in Appendix B.

**Baselines and Implementation Details**. To showcase the efficacy of our proposed Curvature-aware Attention, we equip it with the Graph Transformer and denote the resulting method as Curv-GT. We evaluate Curv-GT against the following neural PDE solvers including, GCN (Xu et al., 2023), GAT (Veličković et al., 2018), DeepONet (Lu et al., 2021; Jin et al., 2022), Graph U-Net (Gao & Ji, 2019), Graph

Transformer (Yun et al., 2019), MKGN (Li et al., 2020c), Galerkin-type Attention (Cao, 2021), GNOT (Hao et al., 2023), GINO (Li et al., 2023b), Geo-FNO (Li et al., 2023a) and Transolver (Wu et al., 2024). The implementation details are provided in Appendix C.

**Main Results.** We first study the performance of different methods for PDEs on the wrinkle manifold—a complex manifold containing both positive, constant, and negative curvatures (shown in Figure 10). To make it more intuitive, the results are shown in the form of a relative loss $\mathcal{L}/\mathcal{L}_{\text{Base}}$ where $\mathcal{L}_{\text{Base}}$ denotes the loss obtained by simply taking the observation $u^{(t)}$ as the prediction at time $t+1$. As shown in Table 1, our proposed Curv-GT consistently achieves the best results over three wrinkle manifold benchmarks. The non-graph-based models like ResNet and DeepONet struggle with the $p$-Laplacian Diffusion equation as they predict with node features and global 3D coordinates, failing to capture features from neighbors and in addition, solely relying on 3D coordinates limits solving the PDEs in $\mathbb{R}^3$ instead of $\mathcal{M}$, which lose the geometry completely. Traditional graph-based models like GT with various different types of attention are not able to perceive the geometric structure since these attentions are all based on node features. Spectral methods like GCN and GINO can yield large errors as they neglect the local spatial structures. Geo-FNO and Transolver give rise to relative small errors among the baseline methods, indicating the usefulness of geometry information. However, the performance gaps between Geo-FNO, Transolver and Curv-GT also show that directly learning the mapping from parameter space to the manifold or merely leveraging the extrinsic geometry is not sufficient for solving PDEs on manifolds. The experiments on more manifolds are available in Appendix D.

**Ablation on Curvature Geometry.** The performances of different methods with and without our proposed curvature-

*Table 2.* Performances of different methods with/without curvature-aware attention for $p$-Laplacian diffusion on the wrinkle manifold.

| Model | Configuration | $L^2(\%)$ |
|-------|---------------|-----------|
| GAT | Non-Curvature-Aware | 8.47±2.82 |
| GAT | Direct Concatenation | 6.92±1.86 |
| GAT | Linear Mapping | 4.57±0.70 |
| GAT | Curvature-Aware | **3.41±0.59** |
| GT | Non-Curvature-Aware | 9.70±4.24 |
| GT | Curvature-Aware | **1.58±0.84** |
| GNOT | Non-Curvature-Aware | 4.39±1.92 |
| GNOT | Curvature-Aware | **0.92±0.31** |

*Table 3.* Repeated experiments on a torus with different subtree partition schemes.

| Subtree amount | | 33.51±1.61 |
|----------------|-----------------|------------|
| Subtree scale | Mean | 30.63±1.47 |
| | Standard error | 1.92±0.18 |
| Loss | $L^2$ loss (%) | 1.69±2.85 |
| | $H^1$ loss (%) | 1.93±2.82 |

aware attention are reported in Table 2, including GAT, GT, and GNOT. It shows that our proposed curvature-aware attention can outperform their non-curvature-aware counterparts by large margins. Besides, Table 2 also presents the performances of GAT by directly concatenating (Direct Concatenation) the point curvatures to node features and aggregating information with a shared linear mapping (Linear Mapping). It can be observed that the results are much worse than our proposed curvature-aware attention.

**Ablation on Multi-Head Attention**. In this experiment, we aim to verify the efficacy of the proposed muli-head curvature-aware attention. Figure 8 presents the performance change against the number of heads $C$ and the maximum depth of subtrees $d$. It demonstrates that 1) the increase in the number of heads can enhance model accuracy and stability, and it limits the model learning capability if its value is too small; 2) the performance drops if $d$ is too large because it cannot capture the local features well with a large depth. In conclusion, a proper depth should be associated with the specific structure of a discrete manifold.

**Training Time Comparison**. In this experiment, we run different neural PDE solvers on elliptic paraboloids in different resolutions. Figure 9 presents the average one-epoch training times of different methods varying against the graph size ($|V|$). It can be observed that our proposed Curv-GAT is still faster than GINO. In particular, the one-epoch training time over a graph with a size of 2500 is around 15 seconds, which can satisfy the practical requirements.

**Stability of Subtree Partitioning**. The subtree partitioning may be very imbalanced due to random selections. Our 100 trials on a 1,024-node torus show stable results, as shown in Table 3.

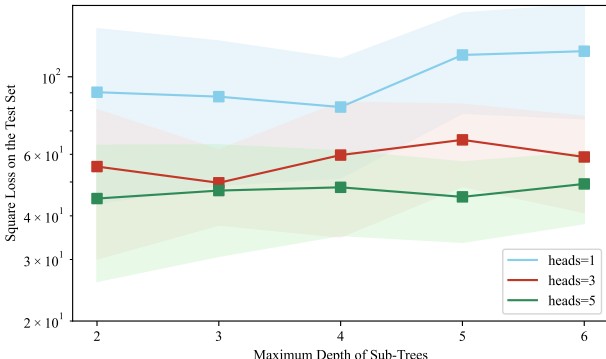

*Figure 8.* Performance change against the number of heads $C$ and the maximum depth of subtrees $d$, the shaded areas around the line indicate standard deviations.

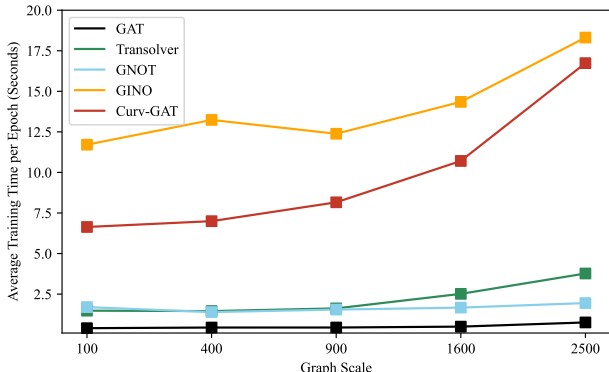

*Figure 9.* Single-epoch training time of different models.

## 6. Conclusion and Future Work

In this paper, a curvature-aware attention is proposed to act as an effective and efficient injected bias for solving PDEs on manifolds. It is achieved via a combination of parallel transport and tensor fields on manifolds. Our proposed curvature-aware attention can be used as a replacement for vanilla attention, and the experimental results show that it can significantly improve the performance of existing methods for solving PDEs on manifolds. In the future, we would like to explore how to extend our curvature-aware attention to address more general tasks on manifolds.

## Acknowledgments

This work is supported by the National Natural Science Foundation of China under Grant No. 62206074 and 62472125, Guangdong Basic and Applied Basic Research Foundation under Grant No. 2025A1515012932, Shenzhen College Stability Support Plan under Grant No. GXWD20220811173233001, Shenzhen Science and Technology Program No. JCYJ20241202123503005 and ZDSYS20230626091203008.

## Impact Statement

We study the neural PDE-solvers by exploring the intrinsic geometry of manifolds. Our work aims to advance scientific computing with machine learning techniques and will not have a negative impact on the community.

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

# A. Formulas on parallel transport

## A.1. Proof of Theorem 4.1

Note that the tangent space at a point on the sphere is isometric to that at any other point. Combined with the fact that parallel transport preserves the inner product. Thus $\mathbf{e}_2$ keeps orthogonal to the tangent vector of the geodesic $\dot{\gamma}$ and its length keeps fixed due to the isometric property. Likewise, $\mathbf{e}_3$ must rotate along with $\gamma(t)$ and the orthogonal property makes it be $\dot{\gamma}$.

To make it self-contained, we offer a complete proof for Theorem 4.2 and Lemma 4.4 whereas the proofs for Lemma 4.3 can be found in (Anderson, 2005),

## A.2. Proof of Theorem 4.2

The metric of $\mathbb{H}^2$ is $g = \frac{1}{Ky^2} dx \otimes dx + \frac{1}{Ky^2} dy \otimes dy$, namely, $g = \mathrm{diag}\{K^{-1}y^{-2}, K^{-1}y^{-2}\}$. Any segment parallel to $y$-axis is a geodesic as said in (Anderson, 2005). One shall compute the Christoffel symbols via Eq. (38) to establish the differential equations of parallel transport.

$$\Gamma^x = \begin{pmatrix} 0 & -\frac{1}{y} \\ -\frac{1}{y} & 0 \end{pmatrix}, \Gamma^y = \begin{pmatrix} \frac{1}{y} & 0 \\ 0 & -\frac{1}{y} \end{pmatrix} \tag{24}$$

One can verify that $U = y\cos\theta\partial_x + y\sin\theta\partial_y$ ($\theta$ is a constant) is indeed a parallel transported vector field by

$$\nabla_{\partial_y} U = 0 \tag{25}$$

## A.3. Proof of Lemma 4.3

The isometry property of Möbius transform group and Poincaré half-plane can be found in (Anderson, 2005), whereas the isometry of translation holds by its definition.

## A.4. Proof of Lemma 4.4

Let $\{x^i\}, \{\tilde{x}^i\}$ denote the coordinate systems on $\mathcal{M}, \mathcal{N}$ and $\varphi : \mathcal{M} \to \mathcal{N}$ be the isomorphic mapping. The corresponding pushforward $\varphi_* : T\mathcal{M} \to T\mathcal{N}$ gives $\varphi_*(\frac{\partial}{\partial x^i}) = \frac{\partial \tilde{x}^j}{\partial x^i} \frac{\partial}{\partial \tilde{x}^j}$.

The isometric condition means $\forall p \in \mathcal{M}, \forall U, V \in T_p\mathcal{M}, g(U,V)|_p = \tilde{g}(\varphi_* U, \varphi_* V)|_{\varphi(p)}$ and equivalently,

$$g_{ij} := g\left(\frac{\partial}{\partial x^i}, \frac{\partial}{\partial x^j}\right) \tag{26}$$

$$= \tilde{g}\left(\varphi_*\left(\frac{\partial}{\partial x^i}\right), \varphi_*\left(\frac{\partial}{\partial x^j}\right)\right) \tag{27}$$

$$= \tilde{g}\left(\frac{\partial \tilde{x}^m}{\partial x^i}\frac{\partial}{\partial \tilde{x}^m}, \frac{\partial \tilde{x}^n}{\partial x^j}\frac{\partial}{\partial \tilde{x}^n}\right) \tag{28}$$

$$= \frac{\partial \tilde{x}^m}{\partial x^i}\frac{\partial \tilde{x}^n}{\partial x^j}\tilde{g}\left(\frac{\partial}{\partial \tilde{x}^m}, \frac{\partial}{\partial \tilde{x}^n}\right) \tag{29}$$

$$=: \frac{\partial \tilde{x}^m}{\partial x^i}\frac{\partial \tilde{x}^n}{\partial x^j}\tilde{g}_{mn} \tag{30}$$

Due to the uniqueness of $\tilde{g}^{jk}$, we can show the identity

$$g^{mn} = \frac{\partial x^m}{\partial \tilde{x}^i}\frac{\partial x^n}{\partial \tilde{x}^j}\tilde{g}^{ij} \tag{31}$$

via direct verification by showing $\tilde{g}^{mn}\tilde{g}_{nl} = \delta_l^m$:

$$g^{mn}g_{nl} = \left(\frac{\partial x^m}{\partial \tilde{x}^i}\frac{\partial x^n}{\partial \tilde{x}^j}\tilde{g}^{ij}\right)\left(\frac{\partial \tilde{x}^p}{\partial x^n}\frac{\partial \tilde{x}^q}{\partial x^l}\tilde{g}_{pq}\right) \tag{32}$$

$$= \frac{\partial x^m}{\partial \tilde{x}^i}\frac{\partial \tilde{x}^q}{\partial x^l}\tilde{g}^{ij}\tilde{g}_{pq}\left(\frac{\partial x^n}{\partial \tilde{x}^j}\frac{\partial \tilde{x}^p}{\partial x^n}\right) \tag{33}$$

$$= \frac{\partial x^m}{\partial \tilde{x}^i}\frac{\partial \tilde{x}^q}{\partial x^l}\tilde{g}^{ij}\tilde{g}_{pq}\left(\frac{\partial \tilde{x}^p}{\partial \tilde{x}^j}\right) \tag{34}$$

$$= \frac{\partial x^m}{\partial \tilde{x}^i}\frac{\partial \tilde{x}^q}{\partial x^l}\tilde{g}^{ij}\tilde{g}_{pq}\delta_j^p \tag{35}$$

$$= \frac{\partial x^m}{\partial \tilde{x}^i}\frac{\partial \tilde{x}^q}{\partial x^l}\delta_q^i \tag{36}$$

$$= \delta_l^m \tag{37}$$

Recall the definition of Christoffel symbol

$$\Gamma_{im}^k := \frac{1}{2}g^{kr}(g_{rm,i} + g_{im,r} - g_{im,r}) \tag{38}$$

and note that

$$g_{rm,i} = \frac{\partial}{\partial x^i}g_{rm} = \frac{\partial \tilde{x}^j}{\partial x^i}\frac{\partial}{\partial \tilde{x}^j}\left(\frac{\partial \tilde{x}^\alpha}{\partial x^r}\frac{\partial \tilde{x}^\beta}{\partial x^m}\tilde{g}_{\alpha\beta}\right) = \frac{\partial \tilde{x}^j}{\partial x^i}\tilde{g}_{\alpha\beta,j}\left(\frac{\partial \tilde{x}^\alpha}{\partial x^r}\frac{\partial \tilde{x}^\beta}{\partial x^m}\right) + \frac{\partial \tilde{x}^j}{\partial x^i}\tilde{g}_{\alpha\beta}\frac{\partial}{\partial \tilde{x}^j}\left(\frac{\partial \tilde{x}^\alpha}{\partial x^r}\frac{\partial \tilde{x}^\beta}{\partial x^m}\right) \tag{39}$$

Substitute $g^{kr}, g_{rm,i}, g_{im,r}, g_{ir,m}$ with the metric tensor in $\mathcal{N}$ and we yield the Christoffel symbol transform formula:

$$\tilde{\Gamma}_{ij}^k = \frac{\partial \tilde{x}^k}{\partial x^\gamma}\left(\Gamma_{\alpha\beta}^\gamma\frac{\partial x^\alpha}{\partial \tilde{x}^i}\frac{\partial x^\beta}{\partial \tilde{x}^j} + \frac{\partial^2 x^\gamma}{\partial \tilde{x}^i\partial \tilde{x}^j}\right) \tag{40}$$

We first show the identity holds for any isometric mapping $\varphi$.

$$\varphi_*(\nabla_X Y) = \tilde{\nabla}_{\varphi_* X}(\varphi_* Y) \tag{41}$$

Let $X = x^i\frac{\partial}{\partial x^i}$ and $Y = y^i\frac{\partial}{\partial x^i}$, the left-hand-side becomes:

$$\varphi_*(\nabla_X Y) = \varphi_*(x^i\nabla_{\frac{\partial}{\partial x^i}}(y^j\frac{\partial}{\partial x^j})) \tag{42}$$

$$= x^i\varphi_*(\frac{\partial y^j}{\partial x^i}\frac{\partial}{\partial x^j} + y^j\Gamma_{ij}^k\frac{\partial}{\partial x^k}) \tag{43}$$

$$= x^i\frac{\partial y^j}{\partial x^i}\varphi_*(\frac{\partial}{\partial x^j}) + x^i y^j\Gamma_{ij}^k\varphi_*(\frac{\partial}{\partial x^k}) \tag{44}$$

$$= x^i\frac{\partial y^j}{\partial x^i}\frac{\partial \tilde{x}^m}{\partial x^j}\frac{\partial}{\partial \tilde{x}^m} + x^i y^j\Gamma_{ij}^k\frac{\partial \tilde{x}^m}{\partial x^k}\frac{\partial}{\partial \tilde{x}^m} \tag{45}$$

$$= x^i\left(\frac{\partial y^j}{\partial x^i}\frac{\partial \tilde{x}^m}{\partial x^j} + y^j\Gamma_{ij}^k\frac{\partial \tilde{x}^m}{\partial x^k}\right)\frac{\partial}{\partial \tilde{x}^m} \tag{46}$$

And the right-hand-side is:

$$\tilde{\nabla}_{\varphi_* X}(\varphi_* Y) = \nabla_{\varphi_*(x^i \frac{\partial}{\partial x^i})}\left(\varphi_*(y^j \frac{\partial}{\partial x^j})\right) \tag{47}$$

$$= x^i \nabla_{\frac{\partial \tilde{x}^l}{\partial x^i} \frac{\partial}{\partial \tilde{x}^l}}\left(y^j \frac{\partial \tilde{x}^k}{\partial x^j} \frac{\partial}{\partial \tilde{x}^k}\right) \tag{48}$$

$$= x^i \frac{\partial \tilde{x}^l}{\partial x^i} \nabla_{\frac{\partial}{\partial \tilde{x}^l}}\left(y^j \frac{\partial \tilde{x}^k}{\partial x^j} \frac{\partial}{\partial \tilde{x}^k}\right) \tag{49}$$

$$= x^i \frac{\partial \tilde{x}^l}{\partial x^i}\left(\frac{\partial y^j}{\partial \tilde{x}^l} \frac{\partial \tilde{x}^k}{\partial x^j} \frac{\partial}{\partial \tilde{x}^k} + y^j \frac{\partial^2 \tilde{x}^k}{\partial \tilde{x}^l \partial x^j} + y^j \frac{\partial \tilde{x}^k}{\partial x^j} \tilde{\Gamma}_{lk}^m \frac{\partial}{\partial \tilde{x}^m}\right) \tag{50}$$

$$= x^i \frac{\partial \tilde{x}^l}{\partial x^i}\left(\frac{\partial y^j}{\partial \tilde{x}^l} \frac{\partial \tilde{x}^k}{\partial x^j} \frac{\partial}{\partial \tilde{x}^k} + y^j \frac{\partial \delta_l^k}{\partial x^j} + y^j \frac{\partial \tilde{x}^k}{\partial x^j} \tilde{\Gamma}_{lk}^m \frac{\partial}{\partial \tilde{x}^m}\right) \tag{51}$$

$$= x^i \frac{\partial \tilde{x}^l}{\partial x^i}\left(\frac{\partial y^j}{\partial \tilde{x}^l} \frac{\partial \tilde{x}^m}{\partial x^j} + y^j \frac{\partial \tilde{x}^k}{\partial x^j} \tilde{\Gamma}_{lk}^m\right) \frac{\partial}{\partial \tilde{x}^m} \tag{52}$$

It suffices to show that

$$\frac{\partial y^j}{\partial x^i} \frac{\partial \tilde{x}^m}{\partial x^j} + y^j \Gamma_{ij}^k \frac{\partial \tilde{x}^m}{\partial x^k} = \frac{\partial \tilde{x}^l}{\partial x^i}\left(\frac{\partial y^j}{\partial \tilde{x}^l} \frac{\partial \tilde{x}^m}{\partial x^j} + y^j \frac{\partial \tilde{x}^k}{\partial x^j} \tilde{\Gamma}_{lk}^m\right) \tag{53}$$

First, note that $\frac{\partial y^j}{\partial \tilde{x}^l} \frac{\partial \tilde{x}^l}{\partial x^i} = \frac{\partial y^j}{\partial x^i}$ and it is thus reduced to show

$$\Gamma_{ij}^k \frac{\partial \tilde{x}^m}{\partial x^k} = \frac{\partial \tilde{x}^l}{\partial x^i} \frac{\partial \tilde{x}^k}{\partial x^j} \tilde{\Gamma}_{lk}^m \tag{54}$$

And the identity holds since

$$\frac{\partial \tilde{x}^l}{\partial x^i} \frac{\partial \tilde{x}^k}{\partial x^j} \tilde{\Gamma}_{lk}^m = \frac{\partial \tilde{x}^l}{\partial x^i} \frac{\partial \tilde{x}^k}{\partial x^j} \frac{\partial \tilde{x}^m}{\partial x^\gamma}\left(\Gamma_{\alpha\beta}^\gamma \frac{\partial x^\alpha}{\partial \tilde{x}^l} \frac{\partial x^\beta}{\partial \tilde{x}^k} + \frac{\partial^2 x^\gamma}{\partial \tilde{x}^l \partial \tilde{x}^k}\right) \tag{55}$$

$$= \frac{\partial \tilde{x}^m}{\partial x^\gamma} \Gamma_{\alpha\beta}^\gamma \left(\frac{\partial \tilde{x}^l}{\partial x^i} \frac{\partial x^\alpha}{\partial \tilde{x}^l}\right)\left(\frac{\partial \tilde{x}^k}{\partial x^j} \frac{\partial x^\beta}{\partial \tilde{x}^k}\right) + \frac{\partial \tilde{x}^k}{\partial x^j} \frac{\partial \tilde{x}^m}{\partial x^\gamma}\left(\frac{\partial \tilde{x}^l}{\partial x^i} \frac{\partial}{\partial \tilde{x}^l}\right)\left(\frac{\partial x^\gamma}{\partial \tilde{x}^k}\right) \tag{56}$$

$$= \frac{\partial \tilde{x}^m}{\partial x^\gamma} \Gamma_{\alpha\beta}^\gamma \delta_i^\alpha \delta_j^\beta + \frac{\partial \tilde{x}^k}{\partial x^j} \frac{\partial \tilde{x}^m}{\partial x^\gamma}\left(\frac{\partial \delta_i^\gamma}{\partial \tilde{x}^k}\right) \tag{57}$$

$$= \frac{\partial \tilde{x}^m}{\partial x^\gamma} \Gamma_{ij}^\gamma \tag{58}$$

$$= \Gamma_{ij}^k \frac{\partial \tilde{x}^m}{\partial x^k} \tag{59}$$

Whence for any parallel transported vector field $U(s)$ along geodesic $\gamma$ parametrized by arc length $s \in [0, T]$ in $\mathcal{M}$, we must have $\nabla_{\dot{\gamma}} U = 0$. Note that the isomorphic mapping $\varphi$ maintains arc length and we can see that $0 = \varphi_*(\nabla_{\dot{\gamma}} U) = \tilde{\nabla}_{\varphi_*(\varphi(\dot{\gamma}))}(\varphi_* U), \forall s \in [0, T]$. Therefore $\varphi_* U$ is also a parallel transported vector field along $\varphi(\gamma)$. Note that

$$\Gamma(\gamma)_0^s U(0) = U(s) = \varphi_*^{-1} \circ \varphi_*(U(s)) = \varphi_*^{-1}(\Gamma(\varphi(\gamma))_0^s \varphi_* U(0)) \tag{60}$$

It is then proved that the pushforward and the parallel transport commute.

### A.5. Proof of Theorem 4.5

Consider the isometric mapping, $\varphi \circ \mathcal{T}_d : z := x + \tilde{y}i \mapsto \tilde{z} = -\frac{1}{z} := \tilde{x} + \tilde{y}i$. And we have:

$$\tilde{x} = -\frac{x}{x^2 + y^2}, \tilde{y} = \frac{y}{x^2 + y^2} \tag{61}$$

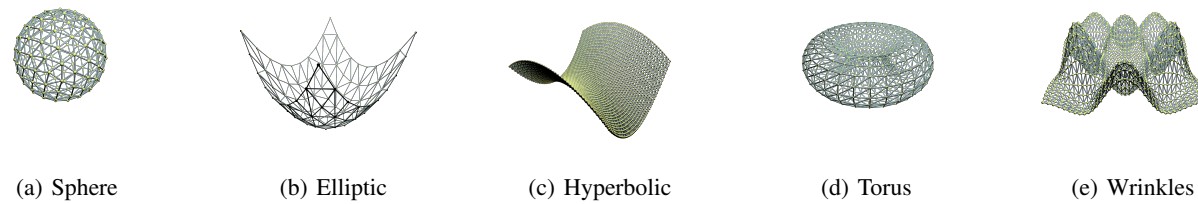

| | (a) Sphere | (b) Elliptic | (c) Hyperbolic | (d) Torus | (e) Wrinkles |

*Figure 10.* The manifolds studied in the datasets.

*Table 4.* Various surfaces are used as background manifolds and PDEs listed in Section B.1 are generated on all of them.

| Surface | Sphere | Elliptic Paraboloid | Hyperbolic Paraboloid | Torus | Wrinkles |
| --- | --- | --- | --- | --- | --- |
| Vertex amount | 162 | 2,500 | 2,500 | 512 | 1,225 |
| Edge amount | 960 | 14,406 | 14,406 | 3,072 | 6,936 |
| Face amount | 320 | 4,802 | 4,802 | 1,024 | 2,312 |
| Train set size | 396 | 396 | 396 | 396 | 396 |
| Test set size | 99 | 99 | 99 | 99 | 99 |

Then $\varphi_* : T_z\mathbb{H}^2 \to T_{\tilde{z}}\mathbb{H}^2$ is:

$$\varphi_*\begin{pmatrix}\frac{\partial}{\partial x}\\\frac{\partial}{\partial y}\end{pmatrix} = \begin{pmatrix}\frac{\partial \tilde{x}}{\partial x} & \frac{\partial \tilde{y}}{\partial x}\\\frac{\partial \tilde{x}}{\partial y} & \frac{\partial \tilde{y}}{\partial y}\end{pmatrix}\begin{pmatrix}\frac{\partial}{\partial \tilde{x}}\\\frac{\partial}{\partial \tilde{y}}\end{pmatrix} \tag{62}$$

$$= \frac{1}{x^2 + y^2}\begin{pmatrix}x^2 - y^2 & 2xy\\-2xy & x^2 - y^2\end{pmatrix}\begin{pmatrix}\frac{\partial}{\partial \tilde{x}}\\\frac{\partial}{\partial \tilde{y}}\end{pmatrix} \tag{63}$$

$$= |z|^{-2}\begin{pmatrix}x^2 - y^2 & 2xy\\-2xy & x^2 - y^2\end{pmatrix}\begin{pmatrix}\frac{\partial}{\partial \tilde{x}}\\\frac{\partial}{\partial \tilde{y}}\end{pmatrix} \tag{64}$$

$$=: \mathbf{J}|_z\begin{pmatrix}\frac{\partial}{\partial \tilde{x}}\\\frac{\partial}{\partial \tilde{y}}\end{pmatrix} \tag{65}$$

Then the pushforward of $\mathcal{T}_{-a} \circ \varphi_* \circ \Gamma(\varphi(\gamma))_0^s \circ \varphi_* \circ \mathcal{T}_a$ is given by $\mathrm{id} \circ \mathbf{J}|_{\tilde{z}_u} \circ \frac{\tilde{y}_u}{\tilde{y}_v} \mathrm{id} \circ \mathbf{J}|_{z_v+a} \circ \mathrm{id}$.

## B. Benchmark Details

### B.1. Time-dependent Equations

Consider the Riemannian manifold $(\mathcal{M}, g)$ and we have the following common useful operators: the gradient operator $\nabla$, the divergence operator $\mathrm{div}$ and the Laplacian-Beltrami operator $\Delta$.

$$\nabla u := g^{ij}\frac{\partial u}{\partial x^i}\frac{\partial}{\partial x^j} \tag{66}$$

$$\mathrm{div}\, Y := |\det g|^{-\frac{1}{2}}\frac{\partial}{\partial x^i}\left(|\det g|^{\frac{1}{2}}y^i\right) \tag{67}$$

$$\Delta u := |\det g|^{-\frac{1}{2}}\frac{\partial}{\partial x^i}\left(|\det g|^{\frac{1}{2}}g^{ij}\frac{\partial u}{\partial x^j}\right) \tag{68}$$

where $X = y^i\frac{\partial}{\partial x^i}$.

**Heat Equation**. The governing equation of a temperature field on the manifold $\mathcal{M}$ is:

$$\frac{\partial u}{\partial t}(x, t) - \Delta u(x, t) = f(x, t) \tag{69}$$

where $u$ is the temperature of $x \in \mathcal{M}$ at time $t$, $\Delta$ is the Laplacian-Beltrami operator associated with $\mathcal{M}$ and $f$ is the source term. The neural operator is required to predict the temperature field at time $t+1$ with observations of $u(x,t)$ and $f(x,t)$ at time $t$.

**Wave Equation**. Wave equations can be used to depict the behaviors of electric field intensity $\mathbf{E}(x,t)$ and magnetic induction $\mathbf{B}(x,t)$ etc in electro dynamics and Schrödinger Equation in quantum theory. The wave equation adopted in the dataset is of the simplified form:

$$\frac{\partial^2 u}{\partial t^2}(x,t) - \Delta u(x,t) = f(x,t) \tag{70}$$

Compared with Eq. (69), it only differs in the time-dependent term. Note that the second-order derivative offers an extra degree of freedom and thus the neural operator is required to predict the field at time $t+1$ given two observations of $u(x,t)$ and $f(x,t)$ at time $t$ and $t-1$.

**Non-Isotropic Diffusion Equations**. The diffusion term in Eq. (69) is linear and isotropic, however, this may fail to reflect the phenomenon in some heterogeneous materials since the diffusion velocity in different directions may differ. The adopted governing PDE is given by:

$$\frac{\partial^2 u}{\partial t^2}(x,t) - \mathrm{div}(a(x)\nabla u(x,t)) = f(x,t) \tag{71}$$

Therefore, the neural operator takes observations of $u(x,t), a(x), f(x,t)$ at time $t$ as inputs and outputs a prediction for $u(x,t)$ at time $t+1$. In some scenarios, $a(x)$ is modeled to be $|\nabla u|^{p-2}$ and the diffusion term is termed as p-Laplacian $\Delta_p$:

$$\Delta_p u := \mathrm{div}(|\nabla u|^{p-2}\nabla u) \tag{72}$$

The associated PDE turns to be:

$$\frac{\partial u}{\partial t}(x,t) - \Delta_p u(x,t) = f(x,t) \tag{73}$$

In a more complicated scenario, for example, a generalized non-Newtonian fluid can be modeled after an anisotropic dissipative potential restricted to two power laws (Ciani et al., 2024), which is:

$$\frac{\partial u}{\partial t}(x,t) - \mathrm{div}\left((|\nabla u|^{p-2} + a(x,t)|\nabla u|^{q-2})\nabla u\right) = f(x,t) \tag{74}$$

## C. Implementation Details

The experiments were conducted on *Ubuntu 20.04 LST* equipped with 4 *NVIDIA RTX A6000 GPU*s, each with 48 GB of GPU memory. Our method is implemented with *Pytorch 1.13* and *Python 3.12*. The initial learning rate $\gamma$ of each model is selected from the set $\{i \times 10^{-j} : i \in \{1,5\}, j \in \{1,2,3\}\}$ to optimize its performance. Besides, the Adam optimizer is used with a decay rate $\beta_1 = 0.9$. Datasets are partitioned into train-set and test-set by ratio 0.8 randomly. A validation set is then created with a ratio of 0.1 from the train-set. The batch size is fixed at either 10 or 20 up to the graph scale. Specifically, for experiments on the sphere, torus and wrinkle manifold, the batch size is set to 20 while 10 is adopted for the others. All models are trained for 300 epochs. The one behaving best on the validation set is selected to participate in the comparison. The curvature threshold $\varepsilon$ is set to $10^{-3}$, in order to both avoid numerical instability and capture the curvature information of most vertices. All curvature-aware models use a 3-head curvature-aware attention mechanism, resulting in the attention head amount $D = 3$. Guided by Figure 8, a proper subtree partition maximum depth $d$ is set to 4. For instance, in that kind of partition, the wrinkle is then decomposed into 27 subtrees.

## D. Experiments on More Manifolds

The performance of different methods for solving PDEs on more manifolds, including the Sphere, Elliptic Paraboloid, Hyperbolic Paraboloid, and Torus. The results are reported in Table 5 and visualized in Figure 12-14. Our proposed Curv-GT can achieve the best results in most cases.

We further conduct an experiment on heat equation on the canonical Stanford Bunny (a complex manifold beyond toy examples), in which the dataset is obtained by finite difference method. The result is shown in Table 6 and visualized in Figure 15.

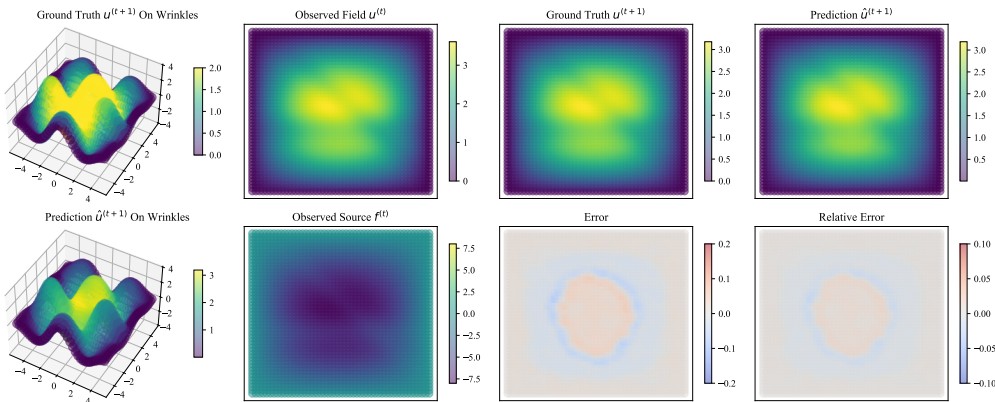

*Figure 11.* Isotropic diffusion on the wrinkle manifold

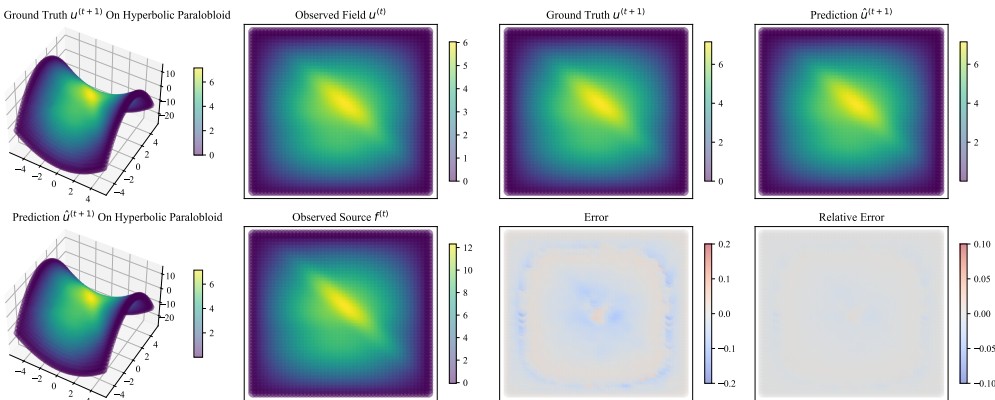

*Figure 12.* $p$-Laplacian on a hyperbolic paraboloid

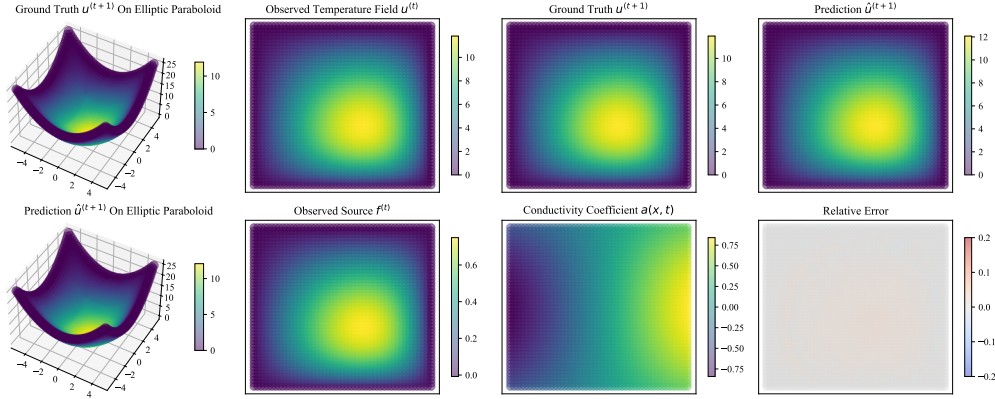

*Figure 13.* Heterogeneous diffusion on an elliptic paraboloid

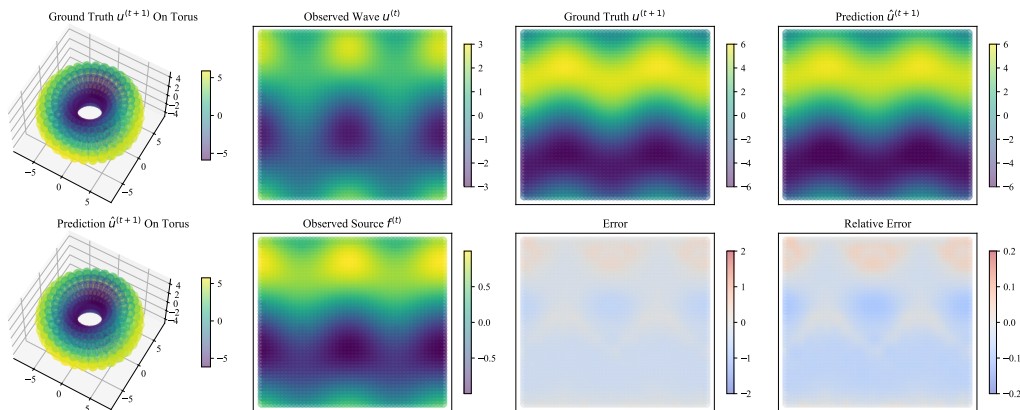

*Figure 14.* Wave equation on a torus

*Table 5.* Performance of different methods on more manifolds.

| Model | Isotropic Diffusion on Elliptic Paraboloid | | Wave Equation on Torus | | Isotropic Diffusion on Sphere | |
|---|---|---|---|---|---|---|
| | $L^2(\%)$ | $H^1(\%)$ | $L^2(\%)$ | $H^1(\%)$ | $L^2(\%)$ | $H^1(\%)$ |
| DeepONet | 19.456±2.168 | 34.418±3.577 | 0.005±0.002 | 0.007±0.002 | 15.929±1.529 | 24.119±3.319 |
| FourierType | 2.399±0.479 | 9.301±1.960 | 0.018±0.004 | 0.025±0.006 | 0.875±0.105 | 1.605±0.151 |
| GalerkinType | 3.082±1.107 | 11.499±4.600 | 0.027±0.013 | 0.043±0.021 | 0.932±0.376 | 1.727±0.562 |
| MKGN | 5.252±0.623 | 27.152±2.998 | 0.120±0.019 | 0.194±0.030 | 5.571±0.268 | 10.135±0.521 |
| GNOT | 0.180±0.064 | 0.345±0.229 | 0.007±0.003 | 0.009±0.003 | **0.060±0.025** | **0.094±0.040** |
| GINO | 19.823±2.626 | 29.597±2.509 | 0.313±0.035 | 0.341±0.037 | 4.898±0.432 | 6.450±0.265 |
| Transolver | 0.099±0.041 | 0.367±0.185 | 0.135±0.043 | 0.269±0.042 | 3.002±0.634 | 4.986±1.155 |
| Curv-GT (**Ours**) | **0.073±0.013** | **0.343±0.048** | **0.002±0.001** | **0.002±0.001** | 0.080±0.011 | 0.131±0.018 |

| Model | $p$-Laplacian Diffusion on Hyperbolic Paraboloid | | Heterogeneous Diffusion on Elliptic Paraboloid | | $p$-Laplacian Diffusion on Torus | |
|---|---|---|---|---|---|---|
| | $L^2(\%)$ | $H^1(\%)$ | $L^2(\%)$ | $H^1(\%)$ | $L^2(\%)$ | $H^1(\%)$ |
| DeepONet | 43.768±1.261 | 53.701±1.683 | 11.623±1.202 | 10.124±1.337 | 64.574±1.695 | 40.440±6.051 |
| FourierType | 15.221±1.761 | 21.972±1.968 | 16.117±1.142 | 16.575±1.943 | 34.360±7.227 | 70.818±1.910 |
| GalerkinType | 16.850±2.951 | 23.587±3.820 | 17.428±1.465 | 16.595±2.513 | 68.499±19.866 | 75.289±17.174 |
| MKGN | 10.649±0.859 | 39.236±2.943 | 3.260±0.628 | 2.658±0.447 | 48.342±8.062 | 58.266±8.042 |
| GNOT | 0.843±0.135 | 1.838±0.341 | 2.228±0.071 | 0.777±0.036 | 4.917±1.440 | 5.445±1.559 |
| GINO | 49.120±0.771 | 46.683±0.918 | 7.078±1.068 | 6.115±1.230 | 58.005±6.218 | 62.090±6.771 |
| Transolver | 1.137±0.192 | 2.152±0.392 | 2.100±0.070 | **0.653±0.067** | 2.903±1.444 | 3.150±1.536 |
| Curv-GT (**Ours**) | **0.799±0.065** | **1.383±0.108** | **1.973±0.149** | 1.677±0.032 | **0.736±0.136** | **1.050±0.130** |

*Table 6.* Different models are compared on the Stanford bunny heat equation dataset.

| Model | Curv-GAT(ours) | GAT | GNOT | Transolver |
|---|---|---|---|---|
| $L^2$ loss (%) | **0.0058±0.0008** | 0.0116±0.0022 | 0.0106±0.0003 | 0.0102±0.0006 |

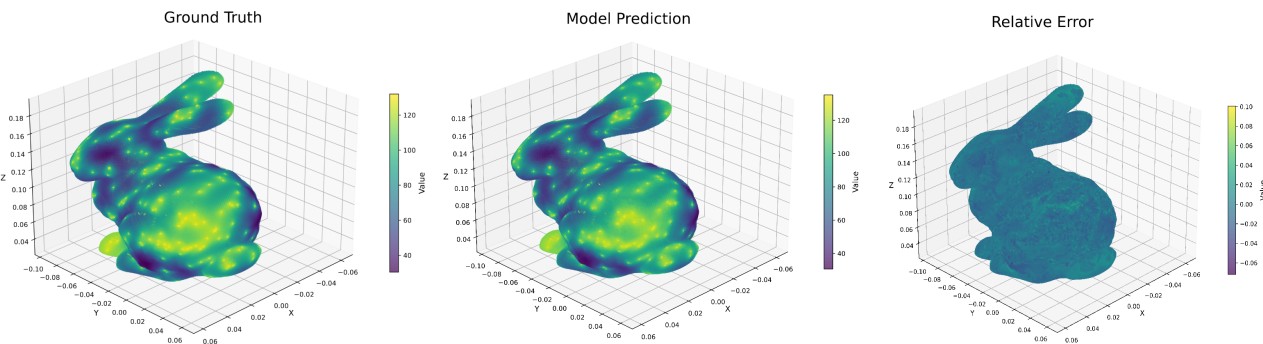

*Figure 15.* Heat equation on Stanford bunny.

