# OpenReview forum: "Curvature-aware Graph Attention for PDEs on Manifolds"
_ICML.cc/2025/Conference — ICML 2025 poster_

### Official Review · Reviewer_DBfv · 2025-03-07

**Overall Recommendation:** 3

**Summary:**

This paper introduces a Curvature-aware Graph Attention method specifically designed for solving PDEs on manifolds. It addresses the limitations of previous approaches that focused on Euclidean spaces or overlooked the intrinsic geometry of manifolds. The proposed method uses fast parallel transport and tensor product on manifolds to fix the original message passing and aggregation process. The authors also introduce a sub-tree partition method to optimize parameter-sharing and reduce computational complexity. Experimental results show this novel attention mechanism improves the performance on solving PDEs on manifolds.

**Claims And Evidence:**

Yes. In the *Experiments* section, the results on solving various PDEs are better than other PDE solvers. In the ablation study, the proposed Curvature-Aware Attention module indeed improves the performance of three attention-based graph neural networks.

**Essential References Not Discussed:**

No, there are no essential related works ignored.

**Experimental Designs Or Analyses:**

For dataset generation, this paper first discretizes the parameterized manifold, then selects a function $u(x,t)$ and computes the sources term $f(x,t)$ to obtain a data tuple $(u^{(t)}, f^{(t)},u^{(t+1)})$. I think this process makes sense.

The main results on Table 1, the authors use 2 criteria $L^2$ and $H^1$, which makes results more convincing and comprehensive.

Although authors have conducted a model on five different 2-manifolds, the Lorentz model for hyperbolic geometry is neglected. I point out this view since the Lorentz model is tightly connected to the theory of relativity, in which the wave equation plays a vital role.

In Appendix H, the authors make visualizations for prediction results, which intuitively support the effectiveness of Curvature-aware Graph Attention.

**Methods And Evaluation Criteria:**

Yes. The evaluation datasets, from three important physical PDEs, are really meaningful.

**Other Comments Or Suggestions:**

1. In *Preliminaries* section, I think there is a typo: “(2,0)-tensor $u^∗ \otimes v^∗$”. (s, t)-tensor $T$ means the multi-linear map  $T$ takes s number of cotangent vectors and t number of tangent vectors as inputs, so this example is about (0,2)-tensor.

**Other Strengths And Weaknesses:**

Pros:
1. Considering the manifold's curvature by parallel transport, instead of explicitly encoding curvature by a neural network.
2. It generalizes the matrix multiplication with the tensor field, combining with the proposed subtree partition to optimize parameter sharing, reducing parameter sizes and computational complexity.
3. The extensive experimental results can support the contribution claimed by the authors, from quantitative and visual experimental results.

Cons:
1. The motivation for using attention-based GNN to solve PDEs is unclear.
2. I think some points need to be discussed further: after running BFS with depth $d$, the scale of subtrees may be very imbalanced, will this situation affect the model performance? Will the random selection for the source nodes affect the results?
3. Eq. (17) contains the parallel transports of tangent vector and covector, but the paper only gives the closed form of tangent vector parallel transport. Under general settings, the two formulas are not the same.
4. For experiments, the wave equation is important for the theory of relativity, which connects tightly with the Lorentz model. Thus, I think this manifold should be considered.

**Questions For Authors:**

1. The method uses local geometry approximation and Gaussian curvature estimation, replacing the complex surfaces locally with constant curvature surfaces. For curvature zero, the surfaces are isomorphic to the Euclidean 2-plane. For positive curvature, the surface is diffeomorphic to a sphere by the Gauss–Bonnet theorem. But for negative curvature, the surfaces are only conformally equivalent to the Poincaré half plane, if a neighborhood on the graph node $u$ comes to large-scale paths or paths around the entire surface, the Poincaré half-plane model may not accurately reflect the parallel transmission characteristics on the original surface. How do you avoid this situation?
2. In the second line of Eq. (17), the authors want to share the parameters from node $u$ to $w$ by parallel transport, but how do you guarantee the consistency between $\Gamma(\eta)_0^t[\tilde{w}_1\otimes \tilde{w}_2]$ and $\tilde{w}_1\otimes \tilde{w}_2$?

**Relation To Broader Scientific Literature:**

The main contribution of this work is that the proposed Curvature-aware Graph Attention replaces vanilla attention on GT and GAT, and it improves the performance on solving PDEs on manifolds. I think this work will affect the methods on solving manifold PDEs, leading them to focus more on intrinsic geometry. Even for the development of graph neural network, this paper may provide some new ideas about geometric GNN.

**Theoretical Claims:**

Yes. I have checked the derivation for the closed form of parallel transport on the sphere and the Poincaré half plane, I think they are correct.

---

> ### Author Rebuttal · Authors · 2025-03-28
>
> # Response To Reviewer DBfv
> We sincerely appreciate your constructive feedback and meticulous evaluation of our work. Below, we provide responses to each point raised.
>
> > **Q1**. In _Preliminaries_ section, there is a typo: (2,0)-tensor $u^∗⊗v^∗$  should be (0,2)-tensor.
>
> **A:** Thank you for catching this typo. We will fix it in the next version of manuscript.
>
> > **Q2.** The motivation for using attention-based GNN to solve PDEs is unclear.
>
> **A:** Reviewer y7yT raised similar concerns. While accuracy is crucial in many applications, **time-sensitive scenarios prioritize speed over minor inaccuracies**, motivating the development of neural solvers like **Neural Operators** in the ML community (see **Related Work: Neural Operator** in our paper).
>
> A key example is **gaming engines**, where faster rendering enhances real-time FPS. To achieve realistic illumination based on blackbody radiation, the heat equation is solved to determine surface temperature distribution. In this case, **the underlying geometry remains fixed**, and high accuracy is less critical as long as the rendering appears plausible. **Smooth scene transitions (speed) matter far more than pixel-perfect details (accuracy).**
>
> Besides, our GNN solver is SE(3)-invariant and can operate without direct coordinate input. Even if retraining is required for a new mesh, **it excels in scenarios where the same mesh appears repeatedly under rotations and translations**, which are common in **material science and gaming engines** (e.g., **triply periodic minimal surfaces** in crystals and block copolymers). In these cases, our method outperforms traditional solvers like FDM and FEM.
>
> We will revise our manuscript to incorporate these examples, further reinforcing the motivation for our approach.
>
> > **Q3.** The subtrees partition may be very imbalanced. Will this affect the model performance? Will the random selection for the source nodes affect the results?
>
> **A:** Your concern about stability makes sense. We conducted an extra experiment involving random subtree partition **100 times** on a torus with **1024 nodes**. It shows that a rather imbalanced partition is unlikely and the outcomes is relatively stable:
> - Subtree amount: 33.51±1.61
> - Mean and standard error of Subtree Scale: 30.63±1.47, 1.92±0.18
> - $L^2$ & $H^1$ loss(%): 1.69±2.85, 1.93±2.82
>
> Other settings align with the original paper.
>
> > **Q4.** Eq. (17) contains the parallel transports(PT) of tangent vector and covector, but the paper only gives the closed form of tangent vector parallel transport.
>
> **A:** By musical isomorphism, we can identify a tangent vector $v$ as a 1-form $v^♭$, a.k.a. a cotangent vector, by $v^♭(u):=\braket{v,u}$. The motivation to introduce PT $Γ(η)$ is to use its **inner-product-preserving nature**. Thus the PT of a 1-form is $Γ(η)v^♭(u):=\braket{Γ(η)v,u}$ and thus is equivalent to PT a tangent vector in implementation.
>
> Thus in your example, the PT of a (0,2)-type tensor is $Γ(η)(\tilde w_1^\*⊗\tilde w_2^\*)(u,v)=\braket{Γ(η)\tilde w_1,u}\braket{Γ(η)\tilde w_2,v}$.
>
>  > **Q5.** For experiments, the wave equation is important for the theory of relativity, which connects tightly with the Lorentz model. Thus, I think this manifold should be considered.
>
> **A:** Yes, the wave equation is crucial in relativity. It will be valuable if our model can also handle it on the Lorentz model, which is, however, not a Riemannian manifold since its metric is **not positive-definite.** This raises questions about whether the tools adopted currently apply to a *psuedo-Riemannian manifold*. This is a constructive suggestion and we will leave it for future work.
>
> From your point of view, I understand that it will be more impactful **if it can handle PDEs in mechanics**. Thus, we also examine the **Navier-Stokes equation** for incompressible and viscous fluid with unit density on a sphere: $∂_t\mathbf u+(\mathbf u·∇)\mathbf u=-∇p+νΔ\mathbf u$. It is insightful in studying the ocean currents on Earth. The experiment setting and dataset generation align with the presented paper.
>
> | Model | $L^2$ loss (%) |
> | --- | --- |
> | Curv-GAT(ours) |**0.128±0.002** |
> | GAT | 1.381±0.006 |
> | Transolver | 3.314±0.651  |
> | GNOT | 0.542±0.164 |
>
> > **Q6.** If a neighborhood has large-scale paths, the Poincaré half-plane model may not accurately reflect the parallel transmission characteristics on the original surface.
>
> **A:** Perhaps our illustration on embedding is also not clear enough to you, you may refer to **Q3 and Q4 in the response to Reviewer y7yT if needed**. We **embed the surface at a node locally** into a constant-curvature surface, with parallel transport decorated **edge-wise, not path-wise**. Subtree partition also mitigates this issue, ensuring consistent parallel transmission unless mesh quality is rather poor.
>
> Thanks again for your energy!

---

### Official Review · Reviewer_Ao56 · 2025-03-09

**Overall Recommendation:** 3

**Summary:**

The authors propose a new PDE-solver based on neural nets for PDE's on manifolds. They claim that taking into account the curvature of the manifold plays a significant role in computing accurately the dynamics of the process to solve.
The authors align the tangent spaces on a manifold via parallel transport use tensors instead of matrix multiplication. This yields  a curvature aware graph attention mechanism, which is better suited for solving PDEs on manifold. The reasoning in sensible. A good literature survey is presented. Performance evaluation is done on various (toy) manifolds and 3 type of PDEs, including nonlinear ones. The methos outperfomrms all methods, in terms of accurace (both L2 and H1), sometimes by an order of magnitude. The paper is supported by theoretical justifications and proofs, along with additional experiments in the appendix. It appears a good fit for the conference.

**Claims And Evidence:**

Claim better solutions on manifolds should take into account curvature consideartation.

**Essential References Not Discussed:**

Refs are fine

**Experimental Designs Or Analyses:**

OK

**Methods And Evaluation Criteria:**

Evalutiona is quite robust, although on toy manifolds, but with increasing complexity, such as Wrinkles. 3 different PDEs are tested, comparison is against 11 competing algorithms, including very recent ones such as Transolver of ICML 2024. The performance of the proposed algorithm consistently outperforms other methods.

**Other Comments Or Suggestions:**

--

**Other Strengths And Weaknesses:**

Writing and illustrations are good.

**Questions For Authors:**

* Time considerations in the computations.
* Have you tried it on more complex manifolds?

**Relation To Broader Scientific Literature:**

Fine

**Theoretical Claims:**

The theory appears fine (although checked only briefly).

---

> ### Author Rebuttal · Authors · 2025-03-27
>
> # Response To Reviewer Ao56
> We are sincerely grateful for the time and effort you have dedicated to reviewing our manuscript. Below, we address each of your comments in detail. Should additional revisions be necessary, we are more than willing to make further adjustments.
>
> > **Q1**. Time considerations in the computations.
>
> **A:** Yes, solving speed is crucial, as it is one of the key advantages of neural solvers over traditional numerical solvers. We find that both the training and inference speeds of our model are acceptable.
>
> We have compared the training times of five neural PDE solvers in Figure 14 (line 973), where our model remains faster than the GNN-based solver GINO. In practice, training time is closely tied to the maximum depth of subtrees $d$ and the number of attention heads. Notably, reducing the number of attention heads to one brings the training speed close to that of Transolver (ICML 2024) and GNOT.
>
> Moreover, once training is complete, neural PDE solvers are much faster than traditional numerical methods in solving forward problems. When it comes to inference speed, there is no significant difference among neural solvers, even on large meshes with around 2,500 vertices. Specifically, as analyzed (line 365), its inference computational complexity is on par with GAT.
>
> > **Q2**.  Have you tried it on more complex manifolds?
>
> **A:** While our current experiments primarily focus on simple manifolds and more complex ones (*wrinkled surfaces*), our framework is designed to generalize to broader geometric settings. In our experiments, we **primarily aim to verify the effectiveness of curvature-awareness via parallel transport** in GNNs, and the results strongly support our assumptions.
>
> Although these manifolds may be considered toy examples, in practice, **many complex surfaces can be decomposed into simpler ones**. These simple manifolds are able to encompass a wide range of common surfaces encountered in game engine design.
>
> We are looking forward to your reply!

---

> > ### Comment · Reviewer_Ao56 · 2025-04-04
> >
> > Thanks for your clarifications. It would have been better to see also a final experiment on more compex data, compared for example with standard known classical numerical methods. However, over all I believe the paper is of interest to the community and maintain my rank.

---

> > > ### Author Response · Authors · 2025-04-05
> > >
> > > Thanks for your recognition and suggestion.
> > >
> > > We further conduct an experiment on heat equation on the canonical **Stanford Bunny (a complex manifold beyond toy examples)**, in which the dataset is obtained by finite difference method. The proposed model is compared with recent baselines listed in our manuscript. The results are presented in the below table and the figures are updated in our anonymous respository at https://anonymous.4open.science/r/icml2025-5376/bunny/bunny.pdf.
> > >
> > > | Model              | $L^2$ Loss (%)    |
> > > | ------------------ | ----------------- |
> > > | **Curv-GAT(ours)** | **0.0058±0.0008** |
> > > | GAT                | 0.0116±0.0022     |
> > > | GNOT               | 0.0106±0.0003     |
> > > | Transolver         | 0.0102±0.0006     |

---

### Official Review · Reviewer_nE4H · 2025-03-10

**Overall Recommendation:** 4

**Summary:**

This paper focus on solving pdes on 2-dim manifolds. It generalizes message passing algorithms to manifolds by adding Gaussian curvature in to consideration. It approximate the complex manifold by constant curvature surfaces in Eq. 11. Such approach using parallel transport on constant curvature surfaces is a better approximation than move vectors in Euclidean spaces. I really like this approach, since constant curvature is one order higher than Euclidean space. The experiments (Fig.8) considered both positive and negative curvatures. Manifold with both positive and negative curvature is also studied.

Weakness: Such approach requires constant curvature. However, it cannot be generalized to high-dim manifolds since in line 139, section curvature will be different that Gaussian curvatures.

In summary, I think considering curvature in solving message passing nn is an interesting task and this work solves this task in 2-dim space. I would rate as accept and I will keep learning this work and update my rating during the discussion period. Looking forward to the reply from the authors!

**Claims And Evidence:**

-

**Essential References Not Discussed:**

-

**Experimental Designs Or Analyses:**

-

**Methods And Evaluation Criteria:**

-

**Other Comments Or Suggestions:**

-

**Other Strengths And Weaknesses:**

-

**Questions For Authors:**

-

**Relation To Broader Scientific Literature:**

-

**Theoretical Claims:**

-

---

> ### Author Rebuttal · Authors · 2025-03-27
>
> # Response To Reviewer nE4H
> We sincerely thank you for your insightful feedback and recognition of our work. We remain fully open to implementing any additional revisions. Below, we address each of your comments in detail.
>
> > **Q1**.  Such approach requires constant curvature.
>
> **A:** This approach only assumes that the surface at a node has approximately constant curvature. Therefore, **the entire surface is not necessary to have constant curvature**, as we solve PDEs on a more complicated manifold *wrinkle* (in Figure 11, line 915). Due to this locality, we cannot directly parallel transport a vector at a node to those too far away since the numerical errors will accumulate. Instead, we turn to a sub-tree partition strategy to balance the trade-off between the computation accuracy and reflecting the curvature changes in a wider range.
>
> > **Q2**. It cannot be generalized to high-dim manifolds since in line 139, section curvature will be different that Gaussian curvatures.
>
> **A:** Your observation here is actually a crucial limitation of the proposed framework. This approach cannot be directly extended to manifolds in higher dimensions since in that case, the **sectional curvature** is no longer a real number.
>
> **Nevertheless, a slight modification is sufficient to extend it to high-dim cases**. If the manifold is compact, then we can use finite charts (each with a coordinate frame) to describe the manifold. Note that sectional curvature is the extended version of Gaussian curvature in higher-dimensional manifolds, and it is indeed a map $K:T\mathcal M\times T\mathcal M\to\mathbb R,(X,Y)\mapsto K(X,Y)$. On a 2d-manifold, it gives Gaussian curvature if the two input tangent vectors are linearly independent.
>
> **This observation sheds light on the high-dim cases.** For instance, on a 3d-manifold with a coordinate frame $\{X_1,X_2,X_3\}$, there are 3 sectional curvatures, $K(X_1,X_2),K(X_2,X_3),K(X_3,X_1)$. We can treat each of them in the same way as the 2D case and combine them. But how to compute them consistently and efficiently requires further consideration. We leave it for our future work.
>
> We will update the manuscript to incorporate the above discussion. Thanks again for your time!

---

> > ### Comment · Reviewer_nE4H · 2025-04-09
> >
> > Dear authors
> >
> > I agree with you on that the entire surface does not have to be constant curvature. Sorry for the misleading words in my initial comment and I didn't mean that. I just meant locally constant curvature.
> >
> > Your comment on Q2 would be an interesting direction. On d-dim spaces, you will need d(d-1)/2 sectional curvatures and a closed-form formula parallel transport can also be challenging. But I think the current manuscript is already good enough for a paper in top conference and good luck for your future research!
> >
> > Best wishes
> >
> > nE4H

---

### Official Review · Reviewer_y7yT · 2025-03-13

**Overall Recommendation:** 2

**Summary:**

The paper proposes a curvature-aware graph attention architecture and applies it to produce a supervised neural time-stepper for PDEs on surfaces embedded in $\mathbb{R}^3$. This architecture leverages the concept of parallel transport on surfaces, and proposes embedding an edge into a constant curvature surface to perform this parallel transport. It builds off of existing frameworks for Graph transformers.

They test their method against many other graph-based network models on the same task, across 4 model geometries and 3 PDEs and show superior performance. Ablation studies also seem to show that the method improves other architectures as well.

## Update after rebuttal:

I appreciated th clarifications from the authors, but I still find the use case for such a method to be relatively niche, and some of the design decisions to be a bit strange. Hence I will keep my score as is (weak reject). If there is sufficient enthusiasm from the other reviewers, I would not stand in the way of ultimate acceptance.

**Claims And Evidence:**

The method takes a well-motivated tack overall, but I did have some more detailed technical questions on the approach, which caused me to question the specifics of the design.

1. I found the embedding into constant-curvature surfaces to be rather coarse. Moreover, it was based off of a curvature estimate at a vertex, but one was philisophically aimed at embedding the edge. Can you explain your reasoning here? If given an explicit triangulation, why not leverage prior works that come up with an explicit notion of discrete connection, e.g., "Globally Optimal Direction Fields" by Knoppel et al.

2. The embeddings themselves seemed a bit unclearly defined to me. For example, in the spherical case, you consider the two edge endpoints as vectors in $\mathbb{S}^2$, but it was not clear how you normalize these and with respect to what center. This would seem to be crucial. (On a related note: it's surprising that one did not just scale the sphere or pseudosphere to account for fractional curvature, as would naturally arise for $K|_u$).

Ultimately, there is no right or wrong on the design choices above, so I should say that on the empirical evidence, I felt that the evaluation and experiments were sufficient for comparison to other graph-based models, and did show significant improved performance.

As for the technical arguments of 4.3, I did not read them carefully, but they seemed correct. I would also like to note that these are not novel in any way, and could have merely referenced existing texts on Riemannian geometry.

**Essential References Not Discussed:**

See above on relation to broader scientific literature.

**Experimental Designs Or Analyses:**

See claims and evidence above.

**Methods And Evaluation Criteria:**

See claims and evidence above.

**Other Comments Or Suggestions:**

Explanation of rating: I gave the paper a borderline reject proposal, due to my uncertainty on the utility of such an approach (see below), and concerns on the technical motivations for the approach (see questions in "Claims and Evidence"). On the positive side, the experiments seem to show significant improvement in the application domain over other methods in its class.

**Other Strengths And Weaknesses:**

None.

**Questions For Authors:**

1. Why would you use such a network to solve a PDE on a surface? I can understand that a trained network is faster than perhaps a more standard numerical solve (like FEM-based). But one must train the network in the first place, presumably with many computed numerical solves. Moreover, any such method is sure to be less accurate and the network is tied to the specific geometry of the surface and would require retraining for any modified geometry.

2. How could this method accomodate node features that cannot be interpreted as elements of the tangent space above a node? It's unclear to me how this could be done, and whether this restriction is vital to the application at hand. In other words, it seems like a well-motivated network model may well take features that cannot be interpreted as elements of the tangent space, so this capability would seem to be very desirable.

**Relation To Broader Scientific Literature:**

The paper is one of several methods aimed at graph-based methods for learning of PDEs and I did not see any glaring omissions amongst the references.

It might be nice to include references to the many recent papers that use a neurally-parameterized space of functions to solve PDEs in an unsupervised fashion, e.g., "Neural Monte Carlo Fluid Simulation" by Jain et al. & "Model reduction for the material point method via an implicit neural representation of the deformation map" by Chen et al.

**Theoretical Claims:**

See claims and evidence above.

---

> ### Author Rebuttal · Authors · 2025-03-28
>
> # Response To Reviewer y7yT
> Thank you for your time and constructive feedback. We appreciate your thorough evaluation and valuable suggestions, which have helped improve our work. Below is our point-by-point response.
>
> >**Q1.**  Why use this GNN for surface PDEs? It is less accurate than FEM. It requires training with precomputed numerical solutions and retraining for any geometry changes due to its surface-specific nature.
>
> **A:** Reviewer DBfv raised similar concerns. Due to space limitations, we kindly refer you to **Q2 in our response to Reviewer DBfv**. We will revise the manuscript to incorporate these examples, further highlighting the motivation behind our proposed method.
>
> > **Q2.** Can it accomodate node features that cannot be interpreted as elements of the tangent space above a node?
>
> **A:** We would like to clarify that our method **can actually** solve PDEs where variables are not tangent vectors. In fact, the PDEs in our experiments involve scalar fields—for instance, in the heat equation, the node feature $u(\mathbf{x})$ represents temperature. Since parallel transport acts on vectors rather than scalars, we **bridge this gap by constructing a natural tangent vector field associated with the function and manifold, namely, the discrete gradient field of temperature (as noted in Remark 2, line 256)**. Furthermore, since the gradient of tensor products can be defined, our approach can theoretically extend to PDEs with tensor variables.
>
> We appreciate this insightful question and will update the manuscript to make it clearer.
>
> > **Q3.** The embeddings themselves seemed a bit unclearly defined to me. For example, in the spherical case, you consider the two edge endpoints as vectors in $S^2$, but it was not clear how you normalize these and with respect to what center. This would seem to be crucial. (On a related note: it's surprising that one did not just scale the sphere or pseudosphere to account for fractional curvature, as would naturally arise for $K|_u$)
>
> **A:** We are sorry for the confusion caused. It would be better to first clarify our embedding methods before resolving your concerns in "Claims and evidence I".
>
> For a node $u$ on a mesh $\mathcal M$, we first compute its Gaussian curvature $K|_u$ at $u$. If $K|_u>ε>0$, then we embed the tangent space $T_u\mathcal M$ at $u$ into a sphere $S$ with curvature $K|_u$ which is tangent to $\mathcal M$ at $u$. Therefore, the edge connecting $u$ and $v$ is embedded into $S$ such that the Euclidean distance between $u,v$ equals the spherical distance (**embedding in isometric sense**). In this way, $\mathcal M$ is locally approximated by $S$ at $u$. Therefore, **we do not need normalizations and we do scale the spheres based on the curvature estimate**. It is the same for pseudospheres.
>
> We will revise the manuscript to further clarify these in our next version.
>
> > **Q4.** It was based off of a curvature estimate at a vertex, but one was philisophically aimed at embedding the edge. Can you explain your reasoning here?
>
> **A:** Sure. It seems embedding at a node $u$ cannot reflect the curvature of an edge $(u,v)$ at first sight. However, in the context of attention mechanism, what matters is how to distinguish the neighbors in a neighborhood. Each $(u,v)$ has a different length and thus the vector will rotate differently with $K|_u>0$. **A feature tangent vector becomes different along different edges due to $K|_u$.** In this sense, the net can discern edge differences based on $K|_u$.
>
> Moreover, under a mild mesh assumption, the geodesic between $u$ and $v$ can be approximated by $K|_u$. This is because, the Jacobi field $J(t)$ has an expansion $|J(t)|^2=t^2-{1\over 3}\braket{R(X,Y)X,Y}t^4+o(t^4)$ where $R$ is the curvature tensor at $u$ and $X,Y\in T_u\mathcal M$. Thus, the edge curvature can be reflected by $K|_u$ if $(u,v)$ is short enough.
>
> > **Q5.**  If given an explicit triangulation, why not leverage prior works that come up with an explicit notion of discrete connection?
>
> **A:** We have read the *Globally Optimal Direction Fields* you suggest and find it less suitable in our model. Discrete connections require geodesic estimation (Eq.(2)), but **as no free lunch in discretization, discrete geodesics must lose some smooth-case properties**. In our implementation, the geodesic we estimate **is obedient to its Euclidean distance**. Besides, **triangulation is not a must in our framework**. So it is just as you say, there is no right or wrong.
>
> > **Q6**. It might be nice to include references to the many recent papers that use a neurally-parameterized space of functions to solve PDEs in an unsupervised fashion.
>
> **A:** Thanks for pointing out the useful references. We will include them in the *Related Work* section in the next version.
>
> Thanks again for your time!

---

### Decision · Program_Chairs · 2025-05-01

**Decision:**

Accept (poster)

**Comment:**

General comment. The authors present a novel approach for operator‑style physical learning by leveraging the geometric properties of the data manifold. They extend self‑attention graph architectures with analytically computed curvature measures based on predefined heuristics. Specifically, they select an appropriate curvature model along the path between nodes u and v and apply a parallel‐transport‑inspired mapping to neighbors’ embeddings during message passing.

The strengths of the work:
- interesting idea of using parallel transport framework to align the representation of the node features on the manifolds\
- interesting idea to apply parallel transport move to the weights of the GNN
- demonstration of the loss improvement when incorporating existing self-attention methods with the proposed approach

Weaknesses
- poor experimental setup - heat equation as well as wave equation are rarely computed on the surface mesh
- absence of comparison with the other methods which use the curvature information - authors compare their method with direct insertion of curvature to the nodes' embedding but there are much more profound method to do so: e.g. https://www.jmlr.org/papers/volume24/23-0064/23-0064.pdf    https://arxiv.org/pdf/2302.08166. Authors only claim Geo-FNO to fail in proper parametrization of complex surfaces but in fact use pretty simple geometries in their experiments.

The reviews were primarily concerned with the following problems:

- how to classify the node based on the local curvature
- overall simplicity of the experimental setup
- problems with the initial graph partition
- inapplicability to the high-dimensional manifolds
- overall motivation to predict the solution of the PDE instead of using a numerical solver

The authors' responses were consistent. In the course of rebuttal they performed experiments on the complex geometry like stanford bunny, and extended the PDE pool by Navier-Stoks equation.

Overall, this work introduces a compelling geometric perspective to graph‑based operator learning. I believe this paper will make a contribution. I encourage the authors to integrate the rebuttal experiments into the final draft. Despite my big concerns about the full scale experiments I won't interrupt the prevailing positive feedback of the work.

Decision: Weak accept.